# Neuronal regulated *ire-1*-dependent mRNA decay controls germline differentiation in *Caenorhabditis elegans*

Mor Levi-Ferber[1], Rewayd Shalash[1], Adrien Le-Thomas[2], Yehuda Salzberg[3], Maor Shurgi[1], Jennifer IC Benichou[1], Avi Ashkenazi[2], Sivan Henis-Korenblit[1]*

[1]The Mina and Everard Goodman Faculty of Life Sciences, Bar-Ilan University, Ramat-Gan, Israel; [2]Cancer Immunology, Genentech, South San Francisco, United States; [3]Department of Neurobiology, Weizmann Institute of Science, Rehovot, Israel

**Abstract** Understanding the molecular events that regulate cell pluripotency versus acquisition of differentiated somatic cell fate is fundamentally important. Studies in *Caenorhabditis elegans* demonstrate that knockout of the germline-specific translation repressor *gld-1* causes germ cells within tumorous gonads to form germline-derived teratoma. Previously we demonstrated that endoplasmic reticulum (ER) stress enhances this phenotype to suppress germline tumor progression(Levi-Ferber et al., 2015). Here, we identify a neuronal circuit that non-autonomously suppresses germline differentiation and show that it communicates with the gonad via the neurotransmitter serotonin to limit somatic differentiation of the tumorous germline. ER stress controls this circuit through regulated inositol requiring enzyme-1 (IRE-1)-dependent mRNA decay of transcripts encoding the neuropeptide FLP-6. Depletion of FLP-6 disrupts the circuit's integrity and hence its ability to prevent somatic-fate acquisition by germline tumor cells. Our findings reveal mechanistically how ER stress enhances ectopic germline differentiation and demonstrate that regulated Ire1-dependent decay can affect animal physiology by controlling a specific neuronal circuit.

**\*For correspondence:**
sivan.korenblit@biu.ac.il

## Introduction

Pluripotency is the developmental potential of a stem cell to give rise to cells of the three embryonic germ layers. Pluripotent stem cells maintain a proliferative, undifferentiated state as long as they reside within a 'niche' that provides continuous pro-mitotic, anti-differentiation cues (*Ohlstein et al., 2004*). Exit from this niche is associated with a tightly regulated switch from self-renewal to differentiation, demonstrating that the immediate microenvironment plays a central role in regulating cell pluripotency. Ultimately, the progressive restriction of cell fate is driven cell autonomously by cell-intrinsic mechanisms that regulate the expression of differentiation promoting genes, including pluripotency-regulatory transcription factors and chromatin remodeling factors, as well as translational control of gene expression (*Schlesinger and Meshorer, 2019*).

The germline stem cells are the ultimate pluripotent cells because they hold the potential to differentiate into all the cell types that comprise the embryo and the adult. Germ cells initially give rise to differentiating gametes, whose somatic fate must be repressed until fertilization. Nevertheless, at a low frequency, germ cells and gametes can generate rare germline tumors containing differentiated somatic cells that are not native to the location in which it occurs (*Ciosk et al., 2006*; *Wright and Ciosk, 2013*). These somatic germline tumors are called teratomas. In humans, the most common form of teratoma originates from oocytes that have entered, but not properly completed, meiosis (*Ulbright, 2005*). The mere existence of ovarian teratomas suggests that oocyte pluripotency is normally restrained to prevent uncontrolled precocious differentiation into somatic cells.

The *Caenorhabditis elegans* gonad provides a well-defined model for studying how germline cells maintain pluripotency in their natural micro- and macroenvironments, that is, in the somatic gonad and in the whole organism (*Seydoux and Braun, 2006*). The *C. elegans* germline stem cells are located in the distal end of the gonad, near the mesenchymal distal tip cell, which interacts with the germ cells and promotes germline stemness (*Byrd and Kimble, 2009*). As in humans, germ cell differentiation is initially limited to meiotic progression and gametogenesis, until the removal of somatic differentiation constraints occurs upon fertilization (*Kuwabara, 2003*). Nevertheless, at a low frequency (enhanced by certain genetic backgrounds), germ cells can abnormally differentiate into many kinds of somatic cells in the absence of fertilization (*Ciosk et al., 2006*; *Biedermann et al., 2009*). We refer to these germline cells, which lose their pluripotency and acquire a differentiated somatic cell fate prior to fertilization, as germline ectopic differentiation (GED). In *C. elegans*, GED can occur by direct conversions of germline cells to somatic cells. Such conversions have been observed upon ectopic expression of a differentiation-promoting transcription factor in the germline (*Tursun et al., 2011*) and upon depletion of the germline-specific translation repressor *gld-1* (*Ciosk et al., 2006*; *Biedermann et al., 2009*). In the absence of *gld-1*, female germline stem cells initiate meiosis, but then exit pachytene and return to the mitotic cycle, yielding a tumorous germline phenotype (*Gumienny et al., 1999*). A fraction of the mitotic germ cells forms a teratoma, which expresses somatic markers and is characterized by aberrantly large, misformed nuclei, readily detectable by DAPI staining (*Ciosk et al., 2006*; *Biedermann et al., 2009*).

The endoplasmic reticulum (ER) mediates correct folding of secretory proteins. ER stress increases the load of misfolded proteins in the ER. In turn, the accumulation of misfolded proteins in the ER activates an ER-adaptive unfolded protein response (UPR). The ER-UPR consists of signaling pathways that help to restore ER homeostasis by reducing the load on the ER and degrading misfolded proteins (*Walter and Ron, 2011*). Previously we have shown that activation of the conserved UPR sensor inositol requiring enzyme-1 (IRE-1) enhances ectopic differentiation of the tumorous germline in *gld-1*-deficient *C. elegans*, which limits the progression of the lethal germline tumor (*Levi-Ferber et al., 2015*). Although the major signaling mode of action of IRE-1 is to generate an active form of the ER stress-related transcription factor *xbp-1* (X-box binding protein-1), ER stress-induced GED requires the *ire-1* gene, but not its downstream target *xbp-1*. The nature of this *ire-1*-dependent *xbp-1*-independent signal remains a mystery (*Levi-Ferber et al., 2015*). *xbp-1*-independent outputs of *ire-1* include activation of signaling cascades by virtue of IRE-1 oligomerization (*Urano et al., 2000*; *Yoneda et al., 2001*), and degradation of ER-localized mRNAs by virtue of IRE-1 ribonuclease activity—a process called regulated Ire1-dependent decay (RIDD) (*Hollien and Weissman, 2006*; *Hollien et al., 2009*). During RIDD, select mRNAs and microRNAs are nicked by IRE1, rendering them vulnerable to rapid degradation by cytoplasmic exoribonucleases (*Hollien and Weissman, 2006*; *Kimmig et al., 2012*; *Guydosh et al., 2017*). Thus far, RIDD has been implicated in a variety of biological processes, including regulation of programmed cell death, cellular differentiation, and inflammation (*Coelho and Domingos, 2014*; *Maurel et al., 2014*; *Lu et al., 2014*).

In this work, we set out to explore how ER stress modulates the germ cell pluripotency/differentiation switch in the tumorous germline of *gld-1*-deficient animals. We discovered that the differentiation of the germline in response to ER stress is regulated at the organismal level and is based on neuronal signaling. We identified a novel ER stress-sensitive neuronal circuit, which communicates with the gonad through the conserved neurotransmitter serotonin to actively repress ectopic germline differentiation. Surprisingly, it is not the stress per se that regulates germ cell fate, but rather the activation of the ER stress response sensor IRE-1 that destabilizes through RIDD the transcripts of a critical neuropeptide implicated in the GED regulatory neuronal circuit. To our knowledge, this is the first example of neuronal circuit inhibition by RIDD.

## Results
### Neuronal IRE-1 regulates ER stress-induced GED

The tumorous germline of *gld-1(-)* animals is prone to undergo ectopic differentiation into somatic cells, which are cleared away from the gonad by programmed cell death followed by corpse absorbance by the surrounding tissues (*Ciosk et al., 2006*; *Levi-Ferber et al., 2015*). These abnormal, ectopically differentiated cells can be identified based on the expression of somatic markers and based

on their relatively enlarged and irregularly shaped nuclei, which are distinct from the typical nuclei of the germline cells. The expression of somatic markers by this subgroup of cells with aberrant nuclei within the gonad of ER-stressed *gld-1(-)* animals has been previously demonstrated, confirming their somatic state (see Figure 3—figure supplement 1 in *Levi-Ferber et al., 2015*). Blocking the ability of the ectopic somatic cells to execute apoptosis increases their half-life in the gonad, facilitating their detection in nearly all of the animals, where they occupy up to 10% of the gonad (*Levi-Ferber et al., 2015*). Previously we have shown that upon exposure to ER stress the population of ectopically differentiated germ cells expands and occupies up to 40% of the gonad of *gld-1(-)* animals, indicating that ER stress promotes the induction of a somatic differentiated state (*Levi-Ferber et al., 2015*). Given the central role of the ER in secretory-protein biosynthesis, including the processing of a variety of hormones and neuropeptides, we wondered whether ER stress-induced GED is a consequence of stress within the germline itself or the perturbation of non-autonomous signals emanating from outside the germline.

We first examined the possibility that ER stress within the germ cells controls GED. To induce ER stress primarily in the germ cells, we used mutants in the *rrf-1* gene, whose RNAi activity is compromised in most somatic tissues but whose germline RNAi activity is intact (*Kumsta and Hansen, 2012*). Specifically, *rrf-1* mutants were treated with a mixture of *gld-1*, *tfg-1*, and *ced-3* RNAi. *gld-1* RNAi induces a germline tumor and sensitizes the germline for GED (*Ciosk et al., 2006*). *tfg-1* RNAi induces ER stress by abrogating protein export from the ER (*Levi-Ferber et al., 2014*; *Witte et al., 2011*). *ced-3* RNAi blocks apoptosis (*Ellis and Horvitz, 1986*) and extends the half-life of the ectopic somatic cells in the gonad, facilitating their detection (*Levi-Ferber et al., 2015*). The efficacy of each of the RNAi treatments was individually confirmed as described in the Materials and methods section. As a proxy for GED, the animals were stained with DAPI, and the presence of ectopically large somatic-like nuclei with irregular shape within the gonad was scored. Strikingly, even though *rrf-1* is not required for RNA interference in the germline, hardly any increase in the amount of ectopic somatic cells in the gonad was apparent upon treatment of *rrf-1* mutants with the *gld-1/tfg-1/ced-3* RNAi (*Figure 1A*). This was in contrast to wild-type animals with intact RNAi machinery, in which approximately 25% of each gonad was occupied by cells with large soma-like nuclei upon exposure to ER stress (*Figure 1A*). This data suggests that ER stress in the soma, rather than only in the germline, drives the induction of aberrant cells within the gonad.

Because IRE-1 is required for ER stress-induced GED (*Levi-Ferber et al., 2015*), we examined whether expression of IRE-1 specifically within the soma is sufficient to drive the induction of ectopic cells within the gonad upon ER stress. To this end, we used *ire-1(-)* mutants, whose germline does not undergo GED in response to ER stress induced by *tfg-1* RNAi treatment (*Levi-Ferber et al., 2015*), and restored IRE-1 expression in its somatic tissues. To restore IRE-1 expression in the entire soma but not the germline, we generated *ire-1(-)* animals expressing an extrachromosomal somatic *ire-1* transgene under its own promoter (note that extrachromosomal arrays are actively silenced in the *C. elegans* germline, but are expressed in the soma; *Kelly et al., 1997*). This line of experiments was carried out in animals treated with *gld-1* and *ced-3* RNAi to facilitate GED detection. As a proxy for GED, the animals were stained with DAPI, and the presence of abnormal nuclei within the gonad was scored. We found that expression of the somatic IRE-1 transgene was sufficient to fully restore the induction of aberrant cells within the gonad upon ER stress (*Figure 1B*).

Next, we examined which somatic tissues mediate ER stress-induced GED. To this end, we restored IRE-1 expression in specific somatic tissues of *ire-1(-)* mutants by driving expression of the rescuing transgene with tissue-specific promoters. We found that expression of *ire-1* under the *rgef-1* pan-neuronal promoter was sufficient to permit ER stress-induced abnormal somatic-like nuclei within the gonad of *ire-1(-)* animals. In contrast, expression of *ire-1* under the *myo-3* muscle promoter or under the *ges-1* intestine promoter did not increase the level of abnormal nuclei in the gonads of ER-stressed animals (*Figure 1B*).

While IRE-1 is normally activated under ER stress conditions, some activation of IRE-1 can be achieved merely by its overexpression (*Kimata et al., 2007*; *Li et al., 2010*; *Salzberg et al., 2017*). Hence, we asked whether increasing IRE-1 expression is sufficient for inducing GED even in the absence of direct ER stress. To this end, we overexpressed *ire-1* transgenes in various tissues of *ire-1(+)*; *gld-1(RNAi)*; *ced-3(RNAi)* animals and followed GED induction in the absence of an ER stress trigger using DAPI staining as a proxy. Overexpression of IRE-1, either in the muscles or in the intestine,

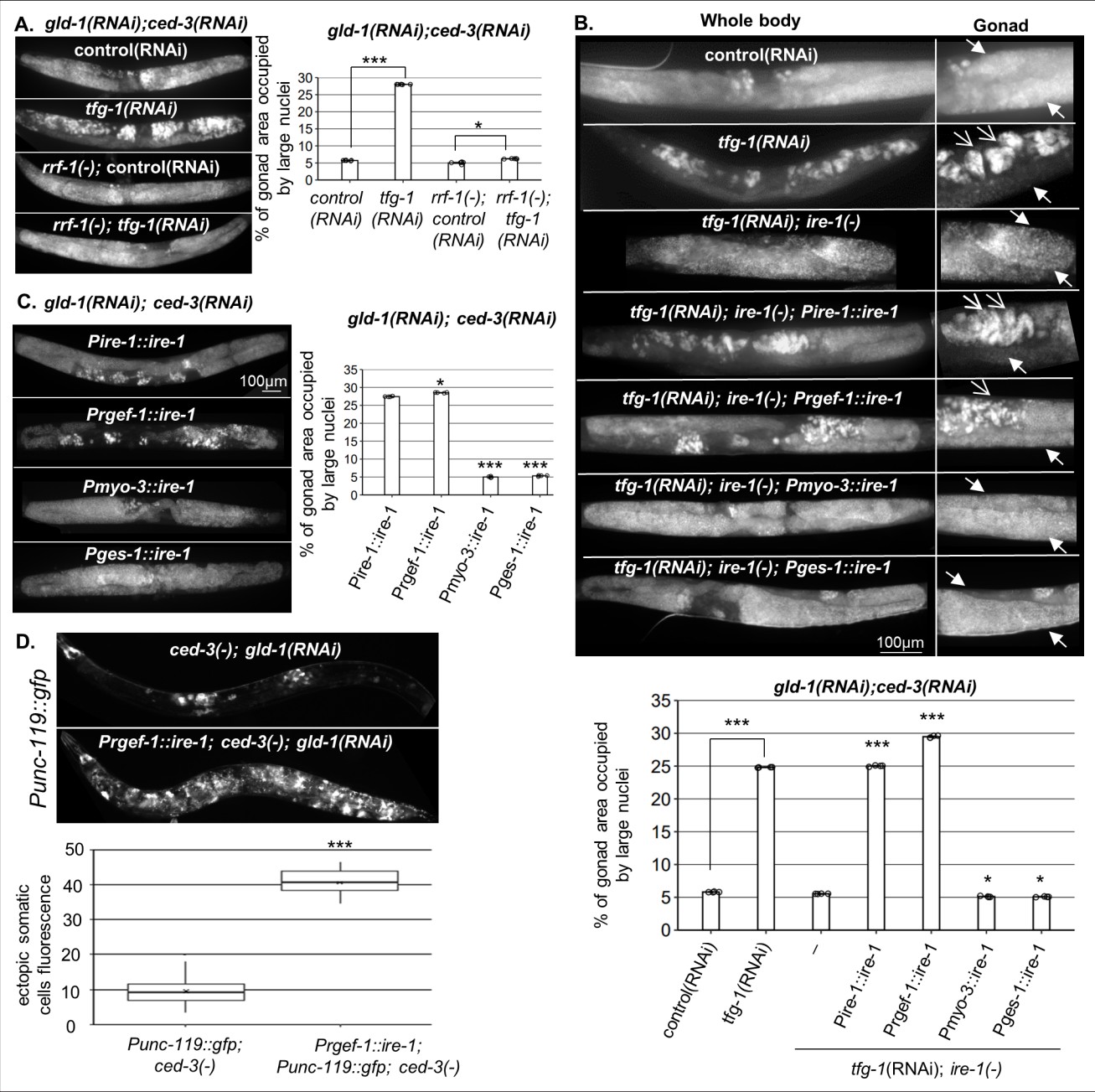

**Figure 1.** Endoplasmic reticulum (ER) stress in the soma regulates germline ectopic differentiation (GED). Percent of gonad area occupied by aberrant somatic-like cells determined by DAPI staining of day 4 *gld-1(RNAi); ced-3(RNAi)* animals. (**A**) *tfg-1* RNAi treatment resulted in increased levels of aberrant somatic-like cells in the gonads of wild-type animals but not in *rrf-1* mutants (n = 320 gonads per genotype, N = 6). Asterisks mark one-way ANOVA values followed by Tukey's post hoc analysis of p<0.001 compared to the same genetic background treated with control RNAi. (**B**) Rescue of *ire-1* expression in the soma (*Pire-1::ire-1*) and in the neurons (*Prgef-1::ire-1*) increased levels of aberrant somatic-like cells in the gonads upon *tfg-1* RNAi treatment, whereas expression of *ire-1* in the muscles (*Pmyo-3::ire-1*) and in the intestine (*Pges-1::ire-1*) did not. Full arrows indicate mitotic germ cells. Open arrows indicate aberrant nuclei (n = 210 gonads per genotype, N = 4). Asterisks mark one-way ANOVA values followed by Tukey's post hoc analysis of p<0.001 compared to *ire-1(-)* animals unless indicated otherwise. (**C**) Overexpression of *ire-1* in the soma (*Pire-1::ire-1*) and in the neurons (*Prgef-1::ire-1*) of animals with wild-type *ire-1* resulted in high levels of aberrant somatic-like cells in the gonads even in the absence of ER stress, whereas overexpression of *ire-1* in the muscles (*Pmyo-3::ire-1*) and in the intestine (*Pges-1::ire-1*) did not (n = 250 gonads per genotype, N = 4). Asterisks mark one-way ANOVA values followed by Tukey's post hoc analysis of p<0.001 compared to *Pges-1::ire-1* animals. (**D**) Overexpression of *ire-1* in the neurons (*Prgef-1::ire-1*) of animals expressing the neuronal reporter *Punc-119::gfp* resulted in high GED levels in *gld-1(RNAi); ced-3(-)* genetic background, even in the absence of an ER stress-inducing treatment. Asterisks mark nested one-way ANOVA values followed by Tukey's post hoc analysis of p<0.001. Means represented by 'X' (n = 70 gonads per genotype, N = 2). Triple asterisks mark significant results of at least a twofold change.

did not lead to the detection of ectopic cells within the gonad of the animals in the absence of ER stress (*Figure 1C*). In contrast, overexpression of IRE-1 in wild-type animals, either in the soma or in the neurons, gave rise to high amount of ectopic somatic-like nuclei even in the absence of ER stress (*Figure 1C*). Furthermore, overexpression of IRE-1 specifically in neurons resulted in expression of the pan-neuronal marker *Punc-119::gfp* throughout the gonads of the transgenic animals, confirming the presence of somatic cells with neuronal characteristics with the gonads of the animals (*Figure 1D*). Since overexpression of IRE-1 in neurons is sufficient for its artificial activation independent of ER stress (*Salzberg et al., 2017*), this data suggests that active IRE-1 signaling in the neurons, rather than neuronal ER stress or ER dysfunction, promotes GED formation in the tumorous gonads of these animals.

## ER stress-induced GED is dictated by specific sensory neurons

Next, we tested whether IRE-1-induced GED is controlled by specific neurons or in a neuron-wide manner. This was examined by expressing *ire-1*-rescuing constructs driven by various neuronal promoters in *ire-1(-); gld-1(RNAi); ced-3(RNAi)* animals and scoring for restoration of ER stress-induced accumulation of ectopic cells in the gonads. We found that rescue of *ire-1* expression in dopaminergic or GABAergic neurons (driven by the *dat-1* and *unc-25* promoters, respectively) did not induce aberrant cells in the gonads under ER stress conditions (*Figure 2A* and *Figure 2—figure supplement 1*). In contrast, rescue of *ire-1* expression in sensory neurons or in glutamatergic neurons (driven by the *che-12* and *eat-4* promoters) led to high levels of aberrant cells in the gonads under ER stress conditions (*Figure 2A* and *Figure 2—figure supplement 1*). Note that although the *che-12* promoter drives expression in 16 sensory neurons, only 10 of them overlap with the *eat-4* promoter (*Supplementary file 1*). Altogether, these findings indicate that the increased generation of aberrant cells upon ER stress is dictated by specific sensory neurons, and it is not a neuron-wide phenomenon. Furthermore, the relevant neurons could be the overlapping subset of sensory and glutamatergic neurons.

## ASI does not regulate GED

Previously, we have shown that ASI regulates ER stress-induced germline apoptosis (*Levi-Ferber et al., 2014*). Furthermore, it is known that sensory information indicating favorable environmental conditions is relayed to the distal tip cell through altered TGFβ production by the ASI neurons (*Dalfó et al., 2012*). ASI is included in the group of sensory neurons covered by the *che-12* promoter. However, it does not communicate with glutamate. Nevertheless, given its known role in regulation of germ cell proliferation and germ cell apoptosis, we examined whether it may also be involved in ER stress-induced GED. Expression of *ire-1* under the *Pdaf-28* promoter (which drives *ire-1* expression in ASI/ASJ neurons) or under *Pdaf-7* (which drives *ire-1* expression specifically in the ASI neuron) did not restore ER stress induction of ectopic cells in the gonads of *gld-1* sensitized *ire-1(-)* animals (*Figure 2A* and *Figure 2—figure supplement 1*). Thus, IRE-1 signaling in ASI is not sufficient for ER stress-induced GED, suggesting that the signaling pathways that regulate ER stress-induced GED and apoptosis are distinct.

## ASE-secreted FLP-6 regulates GED

Next, we hypothesized that the activity of sensory neurons, mediated by the release of glutamate or neuropeptides, communicates a signal that controls germline pluripotency in the gonad. One possibility is that ER stress in the relevant sensory neurons activates a signaling cascade that promotes GED in the gonad. If so, inactivation of the critical signaling molecules/cells of such a germline pluripotency inhibitory/GED-promoting signal should prevent ER stress-induced GED. Alternatively, since ER stress conditions are incompatible with efficient signaling due to secretory defects (*Safra et al., 2013*), ER stress conditions that promote GED formation may do so by interfering with an existing germline pluripotency-promoting signal rather than generating a novel GED-promoting signal. In this case, inactivation of the critical signaling molecules/cells of such a germline pluripotency-promoting pathway should lead to the generation of GED even in the absence of ER stress. To distinguish between these possibilities, we examined whether a deficiency in any of the potential neurotransmitters and neuropeptides, expressed in the 10 glutamatergic sensory neurons, is sufficient to prevent

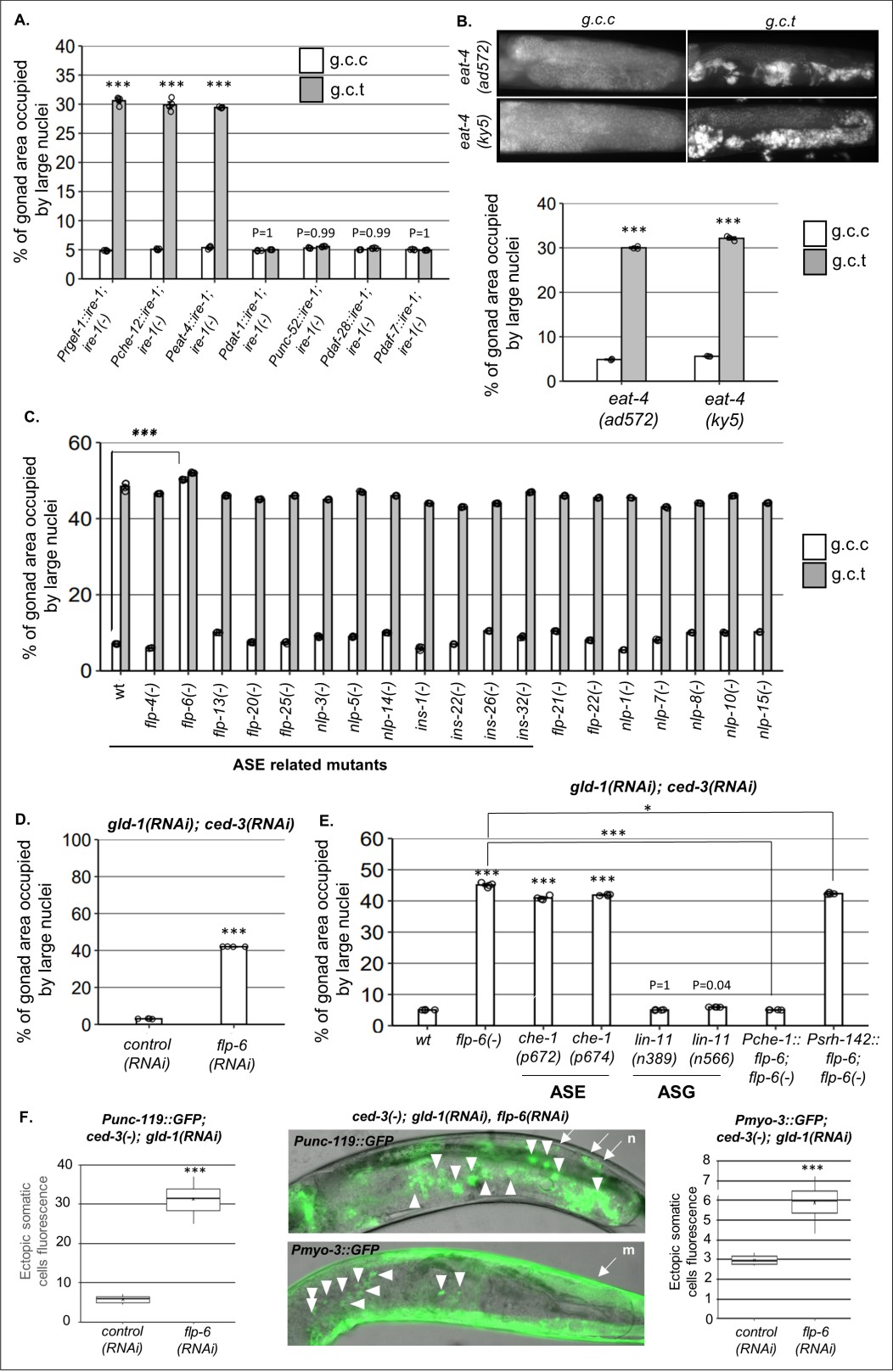

**Figure 2.** ASE-secreted FLP-6 regulates germline ectopic differentiation (GED). Percent of gonad area occupied by ectopic somatic cells determined by DAPI-staining of day 4 animals. *g.c.c* represents treatment with *gld-1, ced-3,* and control RNAi mixture. *g.c.t* represents treatment with *gld-1, ced-3,* and *tfg-1* RNAi mixture. (**A**) Rescue of *ire-1* expression in all neurons (*Prgef-1::ire-1),* in sensory neurons (*Pche-12::ire-1* (or in glutamatergic

*Figure 2 continued on next page*

*Figure 2 continued*

neurons [*Peat-4::ire-1*])) resulted in high levels of ectopic somatic cells in the gonads upon *tfg-1* RNAi treatment, whereas expression of *ire-1* in the dopaminergic (*Pdat-1::ire-1*), GABAergic (*Punc-25::ire-1*), and *ASI/ASJ* neurons (*Pdaf-28::ire-1*and *Pdaf-7::ire-1*) did not (n = 210 gonads per genotype, N = 4). Asterisks mark two-way ANOVA values followed by Tukey's post hoc analysis of p<0.001 relative to the same animals treated with *g.c.c* RNAi. For representative animals, see *Figure 2—figure supplement 1*. (**B**) *eat-4* mutants displayed high levels of ectopic somatic cells in the gonads only upon *tfg-1* RNAi treatment (n = 180 gonads per genotype, N = 3). Asterisks mark two-way ANOVA values followed by Tukey's post hoc analysis of p<0.001 relative to the same animals treated with *g.c.c* RNAi. (**C, D**) *flp-6* mutants (**C**) and *flp-6* RNAi-treated animals (**D**) displayed high levels of ectopic somatic cells in the gonads in the absence of endoplasmic reticulum (ER) stress (n = gonads per genotype, N = 4). Asterisks mark two-way ANOVA followed by Tukey's post hoc analysis of p<0.001 relative to wild-type animals treated with the same RNAi treatment (**C**) and Student's t-test of p<0.001 (**D**). (**E**) *che-1* mutants (with defective ASE) displayed high levels of ectopic somatic cells in the gonads, whereas *lin-11* mutants (with defective ASG) did not. Rescue of *flp-6* expression in ASE (*Pche-1::flp-6*) suppressed GED in *flp-6*(-) mutants, whereas its expression in ADF (*Psrh-142::flp-6*) did not. (n = 195 gonads per genotype, N = 4). Asterisks mark one-way ANOVA values followed by Tukey's post hoc analysis of p<0.001 relative to wild-type animals, unless indicated otherwise. All animals were treated with *gld-1* and *ced-3* RNAi. (**F**) *gld-1(RNAi); ced-3(-)* animals expressing the neuronal reporter *Punc-119::gfp* or the muscle reporter *Pmyo-3::gfp* displayed high GED levels upon treatment with *flp-6* RNAi (n = 70 gonads per genotype, N = 2). Full arrows point at the labeling of somatic neurons (n) or muscles (m) within the soma. Arrowheads point at the labeling of somatic neuron or muscle cells within the gonad. Asterisks mark nested one-way ANOVA values followed by Tukey's post hoc analysis of p<0.001. Means represented by 'X.' Triple asterisks mark significant results of at least a twofold change.

The online version of this article includes the following figure supplement(s) for figure 2:

**Figure supplement 1.** Endoplasmic reticulum (ER) stress-induced germline ectopic differentiation (GED) is controlled by sensory neuronal IRE-1.

---

ER stress-induced GED in ER-stressed animals or sufficient to induce GED in non-stressed *gld-1(RNAi); ced-3(RNAi)* animals.

Since the main neurotransmitter synthesized in the 10 candidate sensory neurons is glutamate, we examined whether glutamate serves as a signaling molecule for GED. To this end, we assessed the level of ectopic nuclei in two different glutamate transport-deficient *eat-4(-)* mutants. Upon treatment with *ced-3* and *gld-1* RNAi, *eat-4(-)* mutants exhibited low levels of abnormal nuclei under non-stress conditions and high levels of abnormal nuclei under ER stress conditions (*Figure 2B*), similarly to wild-type animals (*Figure 1A*). Thus, the interference with glutamate transport does not preclude the induction of GED under ER stress conditions, nor is it sufficient for induction of GED.

We next searched for candidate neuropeptides that may be involved in GED induction. To this end, we analyzed the nuclei pattern in mutants that lack each one of the neuropeptides known to be expressed by the 10 suspected glutamatergic sensory neurons (*Supplementary file 1*). Upon treatment with *ced-3* and *gld-1* RNAi, all of the mutants had high levels of abnormal nuclei in their gonads under ER stress conditions (*Figure 2C*, gray bars). Strikingly, one neuropeptide mutant—*flp-6(ok3056)*—exhibited high levels of abnormal nuclei under non-stress conditions as well (*Figure 2C*, white bars). Similarly high levels of abnormal nuclei under non-stress conditions were obtained by treatment with *flp-6* RNAi (*Figure 2D*). Furthermore, *flp-6* RNAi treatment induced the expression of the *Punc-119::gfp* neuronal marker and the *Pmyo-3::gfp* muscle marker within the gonads of *gld-1(RNAi)*-treated *ced-3(-)*mutants, confirming the presence of somatic cells with neuronal and muscle characteristics with the gonads of the animals (*Figure 2F*). This indicates that the FLP-6 neuropeptide is a negative regulator of GED, consistent with the notion that ER stress may interfere with a fundamental germline pluripotency-promoting signal in the tumorous germline of *gld-1*-deficient animals.

FLP-6 is a secreted neuropeptide with putative hormonal activity, previously implicated in lifespan regulation (*Chen et al., 2016*). It is part of a family of genes encoding FMRF amide-related peptides (FaRPs), which regulate many aspects of behavior (*Li et al., 1999*; *Li and Kim, 2014*). FLP-6 is produced in six neurons (ASE, ASG, AFD, ADF, I1, PVT; *Kim and Li, 2004*). AFD-derived FLP-6 has been implicated in temperature-related longevity regulation (*Chen et al., 2016*). Nevertheless, only ASE and ASG are included in the group of sensory neurons that overlap with the *che-12* and *eat-4* promoters' expression pattern. Therefore, we further assessed ASE and ASG involvement in GED.

Since *flp-6* depletion was sufficient to induce GED in the absence of ER stress, we hypothesized that depletion of its producing cells will have a similar effect. To this end, we genetically inactivated the ASE and ASG cells and examined if this was sufficient to induce GED in *gld-1* and *ced-3 RNAi*-treated animals. We examined two *che-1* mutants, which lack a critical transcription factor involved in ASE fate specification, and thus are ASE deficient (*Sarin et al., 2007*). We observed high levels of ectopic nuclei in the gonads of ASE(-) *che-1* mutants compared to wild-type animals in the absence of ER stress, similar to those observed in *flp-6(-)* mutants (*Figure 2E*). In contrast, two *lin-11* mutants, which lack a critical transcription factor involved in ASG fate specification and are thus ASG defective (*Amon and Gupta, 2017*), exhibited low levels of ectopic nuclei in their gonads, similar to wild-type animals (*Figure 2E*). These observations suggest that the ASE sensory neuron actively prevents GED in *gld-1(RNAi); ced-3(RNAi)* animals.

Taking together the requirement of both ASE and *flp-6* for GED prevention, we hypothesized that ASE-produced FLP-6 suppresses GED in *gld-1(RNAi); ced-3(RNAi)* animals. To examine this, we generated *flp-6(-)* animals expressing a *flp-6* rescuing construct under the ASE-specific promoter *Pche-1*. Unlike *flp-6(-); gld-1(RNAi); ced-3(RNAi)* mutants, which had high levels of abnormal nuclei in their gonads, transgenic animals expressing ASE-produced FLP-6 exhibited low levels of abnormal nuclei in their gonads, similarly to wild-type animals (*Figure 2E*).

Finally, since FLP-6 is a secreted neuropeptide, we wondered whether its secretion by other FLP-6-producing cells may be sufficient for the suppression of GED in *gld-1(-); ced-3(-)* animals. Hence, we examined the levels of GED in *flp-6(-); gld-1(RNAi); ced-3(RNAi)* mutants upon restoration of *flp-6* expression in ADF, one of the six known FLP-6-producing neurons. In contrast to the animals expressing *flp-6* specifically from the ASE neurons, *flp-6(-)* animals expressing *flp-6* rescuing construct under an ADF-specific promoter (*Psrh-142*) exhibited high GED levels similarly to *flp-6(-)* animals (*Figure 2E*). Under the assumption that *flp-6* expression was driven to a similar extent in the two sensory neurons, these results suggest that ASE-produced FLP-6 is a critical signaling moiety in GED regulation in *gld-1(RNAi); ced-3(RNAi)* animals. Furthermore, the fact that FLP-6 production by different sensory neurons leads to distinct phenotypes suggests that FLP-6 may act locally, through synaptic transmission, rather than distally, as a hormone distributed throughout the body cavity of the organism.

## *flp-6* deficiency induces GED independently of ER stress or *ire-1*

Since ER stress promotes GED, we checked whether *flp-6* deficiency is associated with ER stress. First, we examined the possibility that *flp-6* deficiency generates ER stress. To this end, we measured the levels of the IRE-1/UPR reporter *Phsp-4::gfp*, whose transcription is increased under conditions of ER stress in an *xbp-1*-dependent manner. We found that the fluorescence level of the *Phsp-4::gfp* reporter did not increase in animals that lack *flp-6*. Specifically, the level of the IRE-1/UPR reporter was comparable to that of animals with intact *flp-6* when measured throughout the body of the animals (p=0.49, *Figure 3—figure supplement 1A*). Likewise, the level of the *Phsp-4::gfp* reporter measured specifically in the ASE cell body did not increase upon *flp-6* RNAi treatment (in fact, a slight decrease was observed, p=0.0056, *Figure 3—figure supplement 1B*). This data indicates that IRE-1 is not hyperactivated in *flp-6*-deficient animals. To further verify the notion that *flp-6* deficiency-induced GED is not the result of ER stress, we examined if *flp-6* deficiency-induced GED is mediated by *ire-1*, similarly to ER stress-induced GED. We found that *flp-6(-); ire-1(-)* double mutants treated with *gld-1* and *ced-3* RNAi exhibited high levels of abnormal nuclei in their gonads under non-stress conditions, similarly to *flp-6(-); gld-1(RNAi); ced-3(RNAi)* animals (*Figure 3A*). Together, these results show that *flp-6* deficiency can induce GED independently of ER stress or *ire-1*, indicating that *flp-6* acts in parallel or downstream to *ire-1* to regulate GED.

## IRE-1 acts upstream of FLP-6 and controls its transcript abundance via RIDD

GED induction by ER stress in *gld-1(-)* animals is *ire-1*-dependent but *xbp-1*-independent (*Levi-Ferber et al., 2015*). A key *xbp-1*-independent output of *ire-1* is RIDD (*Hollien and Weissman, 2006*), which degrades transcripts based on their proximity to the ER and the presence of a specific hairpin motif (*Oikawa et al., 2010*). Although RIDD is conserved from yeast to human, surprisingly, no RIDD target has been demonstrated thus far in *C. elegans*. We hypothesized that ER stress activates *ire-1*, which in turn degrades *flp-6* transcripts encoding a protein that normally prevents GED. To test this hypothesis,

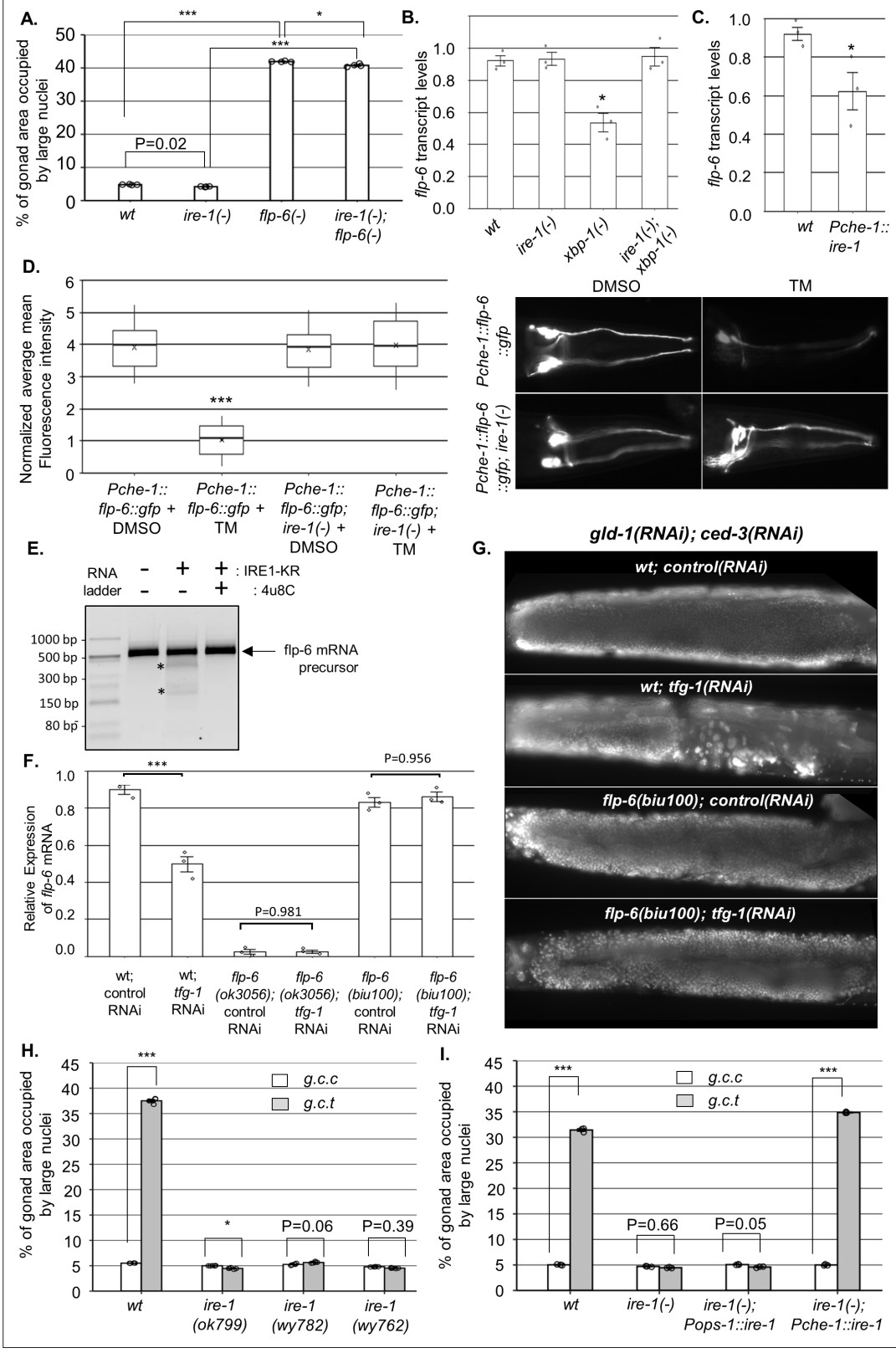

**Figure 3.** IRE-1 acts upstream of *flp-6* and controls its transcript stability. (**A**) *flp-6 ok3056* mutation increased the levels of ectopic somatic cells in the gonads upon treatment with *gld-1 and ced-3* RNAi independently of *ire-1* (n = 200 gonads per genotype, N = 4). The *ire-1 ok799* deletion mutant was examined. Asterisks mark one-way ANOVA values followed by Tukey's post hoc analysis of p<0.001 as indicated. Germline ectopic differentiation

*Figure 3 continued on next page*

*Figure 3 continued*

(GED) levels were assessed by DAPI staining of day 4 animals. Consistent with the lack of dependency on *ire-1* for GED induction, *flp-6* deficiency did not induce expression of the *Phsp-4::gfp* endoplasmic reticulum (ER) stress reporter throughout the body or within the ASE neuron (see ***Figure 3—figure supplement 1***). (**B, C**) *flp-6* transcript levels were assessed by qRT-PCR and normalized to actin transcript levels. *flp-6* transcript levels were significantly reduced in *xbp-1* mutants compared to wild-type animals but not in *ire-1* mutants. *flp-6* transcript levels were also did not significantly change in *ire-1; xbp-1* mutants compared to wild-type animals. Asterisks mark one-way ANOVA values followed by Tukey's post hoc analysis of p<0.001 compared to wild-type animals (**B**). *flp-6* transcript levels were also reduced upon treatment with *tfg-1* RNAi (see ***Figure 3—figure supplement 2A***). *flp-6* transcript levels were significantly reduced in *Pche-1::ire-1* animals, overexpressing *ire-1* in the ASE neurons compared to wild-type animals. Asterisks indicate Student's t-test p-value of <0.05 (**C**). (**D**) L4-staged animals were treated with either 25 µg/ml tunicamycin (TM) or with DMSO till day 2 of adulthood and then scored. TM treatment significantly reduced the fluorescence levels of the translation reporter *Pche-1::flp-6::gfp* compared to DMSO treatment, whereas it did not significantly change the fluorescence levels of *Pche-1::flp-6::gfp; ire-1(-)* animals compared to DMSO. n = 65 gonads per genotype, N = 2. p<0.001 of one-way nested ANOVA followed by Tukey's post hoc analysis is indicated. Means represented by 'X.' *tfg-1* RNAi treatment also reduced the fluorescence levels of the translation reporter *Pche-1::flp-6::gfp* (see ***Figure 3—figure supplement 2C***). In contrast, the fluorescence levels of the *Pflp-6::gfp* and *Pche-1::mcherry* transcriptional reporters were not significantly altered by *tfg-1* RNAi or TM treatment (see ***Figure 3—figure supplement 2B,D,E***). (**E**) Purified recombinant human IRE-1α comprising the kinase and RNase domains (IRE1-KR) was incubated with in vitro-transcribed *flp-6* RNA fragment in the presence of vehicle or 4µ8c (5 µM). Cleavage products of the *flp-6* RNA fragment (marked by asterisks) were observed upon its incubation with IRE1-KR, but not in the presence of the specific IRE-1 ribonuclease inhibitor 4µ8C. Cleavage products of the *flp-6* RNA fragment were not observed upon scrambling the predicted hairpin sequence (see ***Figure 3—figure supplement 3B***). (**F**) *flp-6* transcript levels were assessed by qRT-PCR and normalized to actin transcript levels (N = 3). *flp-6* transcript levels were significantly reduced by *tfg-1* RNAi treatment, but not in *flp-6(biu100)* mutants, in which the putative stem-loop structure has been disrupted while preserving the coding sequence. Negligible levels of *flp-6* transcript were detected in *flp-6(ok3056)* deletion mutants. Asterisks mark nested one-way ANOVA values followed by Tukey's post hoc analysis of p<0.001 compared to the corresponding control RNAi treatment. (**G**) Representative micrographs of whole-body DAPI stained day 4 animals treated with either a mixture of control, *gld-1,* and *ced-3* RNAi or with a mixture of *tfg-1, gld-1,* and *ced-3* RNAi. *tfg-1* RNAi treatment increased the levels of ectopic somatic cells in the gonads of animals with a wild-type *flp-6* transcript (wt) but not in animals with a stem-loop disrupted *flp-6* transcript (*biu100*) (see ***Figure 3—figure supplement 3C*** for quantifications). (**H**) Treatment with a mixture of *gld-1; ced-3; tfg-1* RNAi (*g.c.t*) failed to increase the levels of ectopic somatic cells in the gonads in the absence of functional *ire-1* (n = 210 gonads per genotype, N = 5). *ok799* is an *ire-1* deletion mutation. *wy762* is an IRE-1 endoribonuclease missense mutation. *wy782* is an IRE-1 kinase missense mutation. Asterisks mark two-way ANOVA followed by Tukey's post hoc analysis of p<0.001 as indicated. (**I**) Rescue of *ire-1* expression in the ASE neuron (*Pche-1::ire-1*) restored ER stress-induced GED upon treatment with a mixture of *gld-1; ced-3; tfg-1* RNAi (*g.c.t*), whereas expression of *ire-1* in the ASG neuron (*Pops-1::ire-1*) did not (n = 110 gonads per genotype, N = 3). g.c.c indicates treatment with a mixture of *gld-1; ced-3; control* RNAi. Asterisks mark two-way ANOVA followed by Tukey's post hoc analysis of p<0.001 relative to the same animals treated with *gld-1; ced-3; control* RNAi (*g.c.c*). In all panels, triple asterisks mark significant results resulting in a twofold change or more.

The online version of this article includes the following figure supplement(s) for figure 3:

**Figure supplement 1.** *Phsp-4* expression is not increased by *flp-6* deficiency.

**Figure supplement 2.** *ire-1* regulates *flp-6* transcript levels post-transcriptionally.

**Figure supplement 3.** The *flp-6* hairpin motif is required for ER stress-induced GED in *gld-1*; *ced-3* deficient animals.

---

we measured the mRNA levels of *flp-6* under non-stress and under ER stress conditions using qRT-PCR. Consistent with our hypothesis, ER stress conditions (i.e., *tfg-1* RNAi treatment or mutation in the ER homeostasis-promoting transcription factor *xbp-1*) reduced the levels of the *flp-6* transcript in an *ire-1*-dependent manner (***Figure 3—figure supplement 2A*** and ***Figure 3B***). As expected, deficiency in *ire-1* itself did not reduce *flp-6* transcript levels (***Figure 3—figure supplement 2A*** and ***Figure 3B***). Furthermore, overexpression of *ire-1* specifically in the ASE neuron resulted in a significant reduction in *flp-6* transcript levels relatively to wild-type animals (***Figure 3C***). Together, these data indicate that *flp-6* transcript levels are downregulated by *ire-1* activation in an *xbp-1*-independent manner, as would be expected for a target of RIDD.

The RIDD process destabilizes select RNAs post-transcriptionally. To examine whether the ER stress-related changes in *flp-6* transcript levels are due to transcriptional or post-transcriptional regulation, we first analyzed the expression of a *Pflp-6::gfp* transcriptional reporter. No significant change was observed in the levels of the *flp-6P* transcriptional reporter upon ER stress (i.e., *tfg-1* RNAi treatment) or upon *ire-1* depletion (**Figure 3—figure supplement 2B**). Furthermore, a reduction in the levels of a *flp-6::gfp* translational reporter, under the control of the heterologous ASE-specific *che-1* promoter, was observed upon ER stress (i.e., *tfg-1* RNAi or tunicamycin treatment) in wild-type animals, but not in *ire-1(-)* animals (**Figure 3—figure supplement 2C** and **Figure 3D**). Notably, the reduction in levels of the FLP-6::GFP protein reporter in ER-stressed wild-type animals was not due to a decrease in activity of the driving promoter, which was unaltered by the ER stress treatments (**Figure 3—figure supplement 2D,E**). This data indicates that *ire-1* decreases FLP-6 levels post-transcriptionally, consistent with the hypothesis that *flp-6* transcript is a RIDD substrate.

RIDD substrates typically harbor an *xbp-1*-like stem-loop structure comprising a seven nucleo-tide loop (-C-X-G-C-X-X-X-) with three conserved residues and a stem of at least four base pairs (**Moore and Hollien, 2015**; **Hooks and Griffiths-Jones, 2011**; **Peschek et al., 2015**). Within the spliced *flp-6* sequence, we identified one potential *xbp-1*-like stem-loop structure (**Figure 3—figure supplement 3A**). Although the identified structure contains only six (rather than the typical seven) nucleotides in the loop, it has all the conserved residues (-C-x-G-C-X-X-) followed by a relatively strong putative stem structure. To test whether this RNA molecule could indeed be cleaved directly by the IRE-1 endoribonuclease, we incubated a 513 bp RNA segment harboring this stem loop with a phosphorylated form of purified human IRE-1 kinase-RNase domain (IRE-1 KR-3P) (**Lu et al., 2014**). Indeed, incubation of the *flp-6* stem-loop RNA fragment with recombinant IRE-1 KR-3P generated detectable cleavage products of the expected sizes **Figure 3E**. Furthermore, the cleavage was blocked by the IRE-1-specific RNAse inhibitor 4μ8c (**Cross et al., 2012**; **Figure 3E**) and did not occur upon disruption of the RNA stem-loop structure by nucleotide scrambling (**Figure 3—figure supplement 3A,B**). These data indicate that the *flp-6* RNA can be directly cleaved within its stem-loop structure by IRE-1 RNAse in vitro, thus supporting the possibility that *flp-6* is a direct RIDD substrate.

Next, we used a CRISPR/Cas9 genome engineering protocol to generate *flp-6(biu100)* mutants in which the typical *xbp-1*-like stem-loop structure was destroyed using a set of silent mutations that preserved the coding sequence of the FLP-6 protein (**Figure 3—figure supplement 3A**). We found that unlike the control wild-type animals, the transcript levels of *flp-6* were not decreased by *tfg-1* RNAi-induced ER stress once the stem-loop structure of the transcript was disrupted (**Figure 3F**). Furthermore, we examined how the stem-loop-disrupting mutation affected GED levels in the animals. We found that similarly to wild-type animals with wild-type *flp-6* transcript, treatment with a *gld-1*, *ced-3*, control RNAi mix did not increase the levels of ectopically large nuclei in the gonads of *flp-6(biu100)* mutants (**Figure 3G** and **Figure 3—figure supplement 3C**). This attests to the integrity of the FLP-6 product, whose deficiency results in GED induction (**Figure 2C**). Strikingly, unlike animals with wild-type *flp-6* transcript, treatment with a *gld-1, ced-3, tfg-1* RNAi mix did not induce ectopic large nuclei in the gonads of *flp-6(biu100)* mutants (**Figure 3G** and **Figure 3—figure supplement 3C**). This indicates that the stem-loop motif in the *flp-6* transcript is critical for ER stress-induced GED induction.

Finally, given that it is ASE-produced FLP-6 that is critical for GED prevention, if *flp-6* levels are downregulated by IRE-1's RIDD activity, then the ribonuclease activity of IRE-1 should be essential for ER stress-induced GED. Furthermore, IRE-1 expression specifically in the ASE sensory neuron should be sufficient for restoration of ER stress-induced GED in otherwise *ire-1*-deficient mutants. Indeed, analysis of a kinase-dead *(wy782)* and a ribonuclease-dead *(wy762) ire-1* mutant (**Wei et al., 2015**) indicated that both the kinase and ribonuclease activities of IRE-1 were required for ER-stress induction of GED (**Figure 3H**). Note that the kinase activity is required for the ribonuclease activity, which directly executes RIDD (**Harnoss et al., 2019**). Furthermore, we individually restored *ire-1* expression in ASE or ASG (two FLP-6-producing glutamatergic sensory neurons). As expected, *Pche-1::ire-1; ire-1(-)* transgenic animals, which express *ire-1* specifically in the ASE neuron and consequently destabilize the *flp-6* transcript (**Figure 3C**), had high levels of abnormal nuclei in their gonads upon exposure to ER stress. In contrast, *Pops-1::ire-1; ire-1(-)* transgenic animals, which express *ire-1* in the ASG and ADL sensory neurons, did not have high levels of abnormal nuclei in their gonads upon exposure to

ER stress (**Figure 3I**). Altogether, these findings further support the conclusion that IRE-1 limits FLP-6 protein levels in the ASE neuron by destabilizing *flp-6* transcripts via RIDD.

## AIY prevents GED by acetylcholine (ACh) signaling

ASE is one of the head amphid neurons, and it does not synapse with the gonad. Hence, we examined the putative involvement of four interneurons (AIA, AIB, AIY, RIA) known to synapse with ASE (**Chen et al., 2006**) in the GED process. Our data support the hypothesis that ER stress conditions promoting GED formation interfere with an existing germline pluripotency-inducing signal. Accordingly, inactivation/interference of the critical signaling cells should also lead to the generation of GED, even in

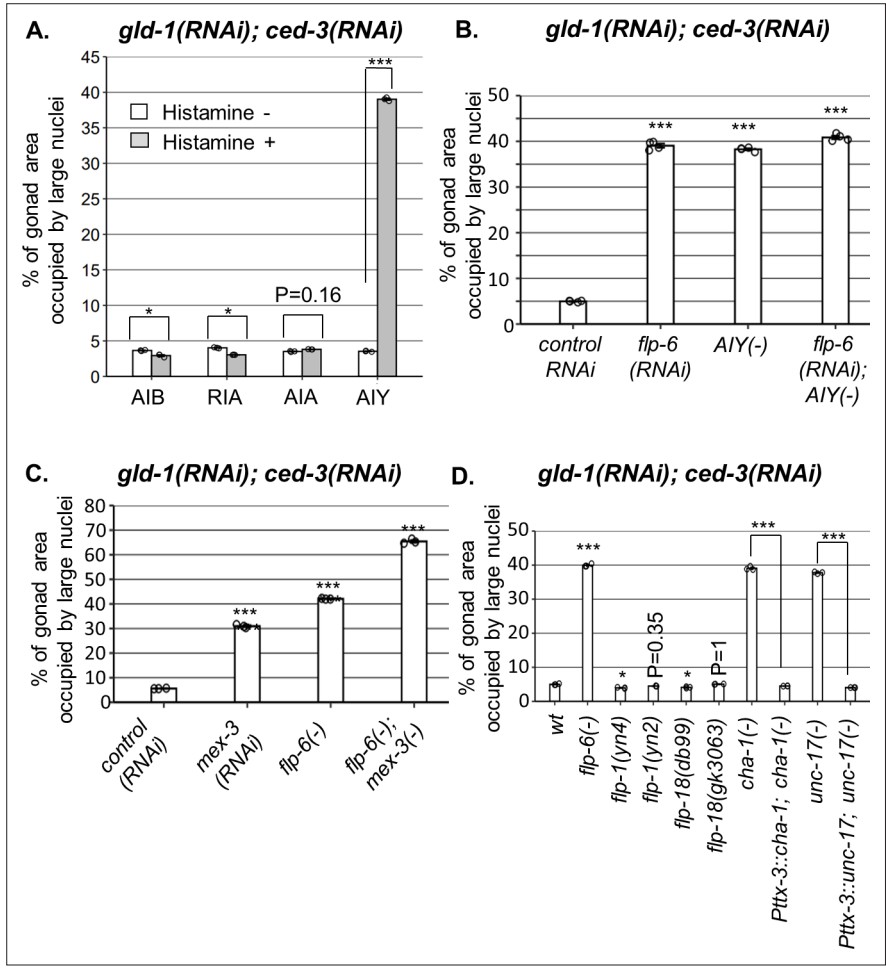

**Figure 4.** AIY produces acetylcholine (ACh) to prevent germline ectopic differentiation (GED). Percent of gonad area occupied by ectopic somatic cells determined by DAPI staining of day 4 animals treated with a mixture of *gld-1* and *ced-3* RNAi. (**A**) Four histamine-inducible interneurons silencing mutants (AIA, AIB, AIY, RIA) were examined (n = 170 gonads per genotype, N = 3). While AIA, AIB, and RIA histamine-treated animals exhibited low levels of ectopic somatic cells in the gonads, AIY histamine-treated animals had high GED levels. Asterisks mark two-way ANOVA followed by Tukey's post hoc analysis of p<0.001 relative to the same animals without histamine treatment. (**B**) *flp-6* RNAi did not additively increase GED levels in AIY histamine-treated animals (n = 180 gonads per genotype, N = 4). p-Values were determined by one-way ANOVA followed by Tukey's post hoc analysis. All strains were grown in the presence of histamine. (**C**) Animals deficient in both *flp-6(ok3056)* and *mex-3* additively increased GED (n = 215 gonads per genotype, N = 3). Asterisks mark one-way ANOVA followed by Tukey's post hoc analysis of p<0.001 relative to animals treated with the *gld-1, ced-3, control* RNAi mix. (**D**) *cha-1(p1152)* and *unc-17(e113)* mutants had high levels of ectopic somatic cells upon *gld-1* and *ced-3* RNAi treatment (n = 190 gonads per genotype, N = 3). These were suppressed upon expression of the corresponding transgenes in the AIY neuron (*Pttx-3::cha-1* and *Pttx-3::unc-17*). Asterisks mark one-way ANOVA followed by Tukey's post hoc analysis of p<0.001 relative to wild-type animals unless indicated otherwise. In all panels, triple asterisks mark significant results resulting in a twofold change or more.

the absence of ER stress, as seen in ASE(-)/*flp-6*(-) animals (*Figure 2D–F*). Hence, we used a system for inducible silencing of specific neurons based on transgenic nematodes engineered to produce the inhibitory *Drosophila* histamine-gated chloride channel (HisCl1) in each of the four suspected interneurons (AIA, AIB, AIY, RIA; *Pokala et al., 2014*). Presence of aberrant nuclei in the gonads of the animals was assessed in each of the strains, with or without histamine, upon treatment with *gld-1* and *ced-3* RNAi. Whereas histamine-induced inactivation of AIA, AIB, and RIA resulted in low levels of abnormal nuclei in the gonads, animals in which the AIY neuron was inactivated displayed high levels of abnormal nuclei in their gonads (*Figure 4A*). This result implicates AIY in GED regulation.

In order to see if ASE-produced *flp-6* and AIY act in a common pathway to regulate GED, we compared the levels of abnormal nuclei between *flp-6* and AIY-defective animals in a *gld-1(-); ced-3(-)* genetic background. We found that each perturbation, individually or in combination, resulted in occupation of ~40% of the gonad with abnormal nuclei (*Figure 4B*). In contrast, treatment of *flp-6* mutants with *mex-3* RNAi (previously reported to increase GED in *gld-1(-)* animals; *Ciosk et al., 2006*) did increase the occupancy of the gonad with GED beyond 40% in an additive manner (*Figure 4C*), demonstrating that the analysis of GED levels in this system is not limited by saturation. Together, these results suggest that the AIY interneuron and *flp-6* act in a common pathway to control GED, whereas *mex-3* deficiency promotes GED via an independent pathway.

We next investigated how the signal is relayed from AIY to the gonad. AIY communicates mainly with *flp-1*, *flp-18* or ACh. However, neither deficiencies in *flp-1* nor in *flp-18* resulted in high levels of abnormal nuclei in the gonads (*Figure 4D*), suggesting that these neuropeptides are not individually necessary for GED. Next, we examined ACh involvement in this signaling pathway using mutations that partially perturb ACh synthesis. Upon treatment with *gld-1* and *ced-3* RNAi, we found high levels of abnormal nuclei in the gonads of *cha-1(p1152)* mutants, in which Ach synthesis is defective (*Cohen et al., 2014*), as well as in *unc-17(e113)* mutants, in which synaptic vesicle Ach transport is defective (*Zhu et al., 2001*; *Figure 4D*). Furthermore, expression of a rescuing construct of *cha-1* or *unc-17* under the AIY-specific *Pttx-3*^promB promoter (*Altun-Gultekin et al., 2001*) led to low levels of abnormal cells in the gonads, similarly to the wild-type level (*Figure 4D*). Altogether, these findings implicate ACh signaling by AIY in the regulation of GED.

## HSN produces serotonin to prevent GED

How is the signaling cascade transmitted downstream to the gonad? AIY communicates via synapses with a few sensory neurons (AWA, AWC, ADF) and a few interneurons (RIA, RIB, AIZ). In addition, AIY synapses with the HSN-R motor neuron, which innervates the vulva muscles (*Chen et al., 2006*). Based on its anatomical location near the vulva, HSN was the strongest candidate for further examination. Indeed, high levels of abnormal nuclei were identified upon treatment with *gld-1* and *ced-3* RNAi in two HSN-deficient mutants (*her-1* and *sem-4*) as well as in three HSN migration-defective mutants (*Figure 5A*). In addition, an epistasis analysis between *flp-6* and *sem-4* or *her-1* HSN-deficient animals showed no additive effect on the levels of abnormal nuclei in the gonad (*Figure 5A*). Together, these results suggest that HSN and *flp-6* act in a common pathway to regulate GED.

How does HSN signal to the gonad? We examined the involvement of four HSN-produced neuropeptides (*nlp-3, nlp-8, nlp-15, flp-5*) and the neurotransmitter serotonin. None of the mutants of the HSN-related neuropeptides induced high levels of GED upon treatment with *gld-1; ced-3* RNAi (*Figure 5B*). In contrast, two serotonin-defective mutants (*tph-1* and *bas-1*) did exhibit high levels of abnormal nuclei in their gonads, similarly to *flp-6(-)* (*Figure 5C*). This indicates that serotonin production is required for preventing the appearance of cells with aberrant nuclei in the gonads of *gld-1(-); ced-3(-)* animals.

There are four genes encoding major serotonin receptors in *C. elegans*. Hence, we examined whether a deficiency in any of these receptors results in GED formation in *gld-1(-); ced-3(-)* animals, similarly to animals deficient in serotonin production. However, we found that GED levels in mutants lacking each of the major serotonin receptors individually (*ser-1(-), ser-4(-), ser-7(-), mod-1(-)* in *gld-1(RNAi); ced-3(RNAi)* background) remained low (*Figure 5—figure supplement 1*). This implies that in this case serotonin affects germ cell fate by another serotonin receptor or that it is redundantly recognized by more than one of these receptors.

Next we examined whether serotonin can suppress GED formation in *gld-1(RNAi); ced-3(RNAi)* background. First, we examined how externally supplied serotonin affected GED by measuring ectopic

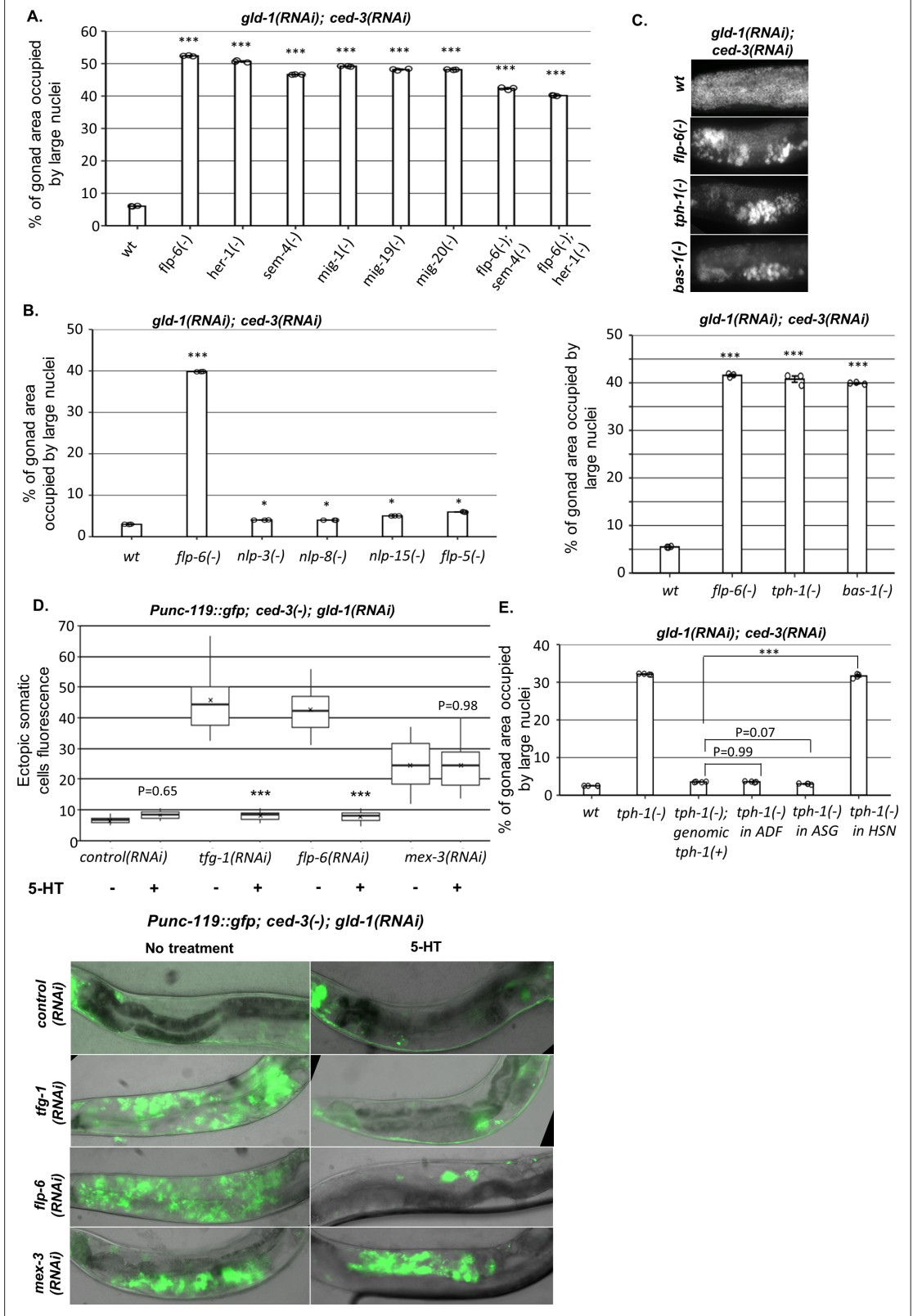

**Figure 5.** HSN produces serotonin to prevent germline ectopic differentiation (GED). (**A–C**) Gonad area occupied by ectopic somatic cells determined by DAPI staining of day 4 animals treated with a mixture of *gld-1* and *ced-3* RNAi. All HSN deficiency and migration mutants had high levels of ectopic somatic cells in their gonads. *flp-6* deletion did not further increase GED levels in HSN-defective animals (n = 175 gonads per genotype, N = 3) (**A**). HSN-related neuropeptide mutants had low levels of GED, similar to wild-type animals (n = 200 gonads per genotype, N = 3) (**B**). Both *tph-1(mg280)*

*Figure 5 continued on next page*

*Figure 5 continued*

and *bas-1(tm351)* serotonin mutants had high levels of ectopic somatic cells in their gonads, similarly to *flp-6* mutants (n = 185 gonads per genotype, N = 3) (**C**). See *Figure 5—figure supplement 1* for the analysis of serotonin receptor mutants. (**D**) Treatment with 20 mM serotonin (5-HT) suppressed the high ectopic somatic fluorescence of the neuronal reporter *Punc-119::gfp* in the gonads of *gld-1(RNAi); tfg-1(RNAi)* or *gld-1(RNAi); flp-6(RNAi)*-treated *ced-3* mutants to the same extent as in the control RNAi group. However, treatment with serotonin (5-HT) did not suppress the ectopic somatic fluorescence of the neuronal reporter in *gld-1(RNAi); mex-3(RNAi)*-treated animals. Representative fluorescent images are shown. n = 90 gonads per genotype, N = 2. Asterisks mark nested one-way ANOVA followed by Tukey's post hoc analysis of p<0.001 relative to the same animals without serotonin supplementation. Means represented by 'X.' (**E**) *tph-1* deficiency in HSN, but not in other serotonin-producing neurons, resulted in high levels of ectopic somatic-like cells in the gonads (n = 195 gonads per genotype, N = 4). In (**A–C**) and (**E**), asterisks mark one-way ANOVA followed by Tukey's post hoc analysis of p<0.001 relative to wild-type animals treated with *gld-1* and *ced-3* RNAi, unless indicated otherwise. In all panels, triple asterisks mark significant results resulting in a twofold change or more.

The online version of this article includes the following figure supplement(s) for figure 5:

**Figure supplement 1.** Four major serotonin receptor mutants are not required for germline ectopic differentiation (GED) in *gld-1; ced-3*-deficient animals.

**Figure supplement 2.** Treatment with 20 mM serotonin (5-HT) decreased the levels of abnormal nuclei in the gonads of *gld-1* and *ced-3* RNAi-treated *tph-1(mg280)* and *flp-6(ok3056)* mutants (n = 165 gonads per genotype, N = 3).

somatic cells fluorescence of animals expressing the neuronal reporter *Punc-119::gfp*. Consistent with the putative role of serotonin in the suppression of GED, we found that supplementation with 20 mM serotonin suppressed the detection of cells expressing the neuronal marker in the gonads of *tfg-1-* or *flp-6*-deficient animals under *gld-1* and *ced-3 RNAi* conditions (*Figure 5D*). In contrast, in *mex-3-*deficient background, serotonin supplementation did not alter the higher ectopic somatic cells' fluorescence levels (*Figure 5D*). Furthermore, treatment with 20 mM serotonin also reduced the levels of abnormal nuclei in the gonads of *flp-6(-)* mutants treated with *gld-1* and *ced-3 RNAi* down to wild-type level (*Figure 5—figure supplement 2*). Similar results were obtained in *tph-1*-deficient animals treated with *gld-1* and *ced-3 RNAi* upon serotonin supplementation (*Figure 5—figure supplement 2*). This suggests that high levels of serotonin are sufficient to suppress GED induced by *flp-6* deficiency in *gld-1(-); ced-3(-)* animals.

Serotonin is produced by the ADF, ASG, and HSN neurons. The fact that HSN-defective *gld-1(-); ced-3(-)* animals have high levels of GED suggested that serotonin production by HSN may be critical for GED suppression. To verify that serotonin production specifically from HSN regulates GED, we measured GED levels upon *gld-1; ced-3* RNAi treatment in strains expressing Cre-Lox-specific knockout of *tph-1* (*Flavell et al., 2013*). Inhibition of serotonin production in either ADF or in ASG resulted in low levels of abnormal nuclei in the gonads, similarly to those of the genomic rescue of *tph-1* (*Figure 5E*). In contrast, inhibition of serotonin production in HSN led to high levels of abnormal nuclei in the gonads, similar to those observed in *tph-1(mg280)* mutants (*Figure 5E*). Altogether, our results suggest that HSN determines GED levels in the tumorous germline of *C. elegans* through serotonin signaling, and that serotonin seems to regulate GED specifically in the context of the ASE-AIY-HSN-germline circuit.

## Perturbation of the ASE-AIY-HSN-gonad circuit induces GED

Thus far, our findings demonstrate that the ASE, AIY, and HSN neurons are required individually to prevent the accumulation of cells with abnormal nuclei in the tumorous gonad of *gld-1*-deficient animals. Previously we have shown that ER stress also results in the accumulation of cells with abnormal nuclei in the tumorous gonad of *gld-1*-deficient animals, and that these cells are ectopically differentiated somatic cells that arise from the tumorous germline (*Levi-Ferber et al., 2015*). Our new findings suggest that ER stress affects the differentiation state of the tumorous germline by compromising the pluripotency-protective ASE-AIY-HSN-germline circuit. Thus, the abnormal nuclei that arise and accumulate in the tumorous gonads upon perturbations within the ASE-AIY-HSN-germline circuit are likely to be the nuclei of differentiated somatic cells as well. To test this directly, we made use of a *Punc-119::gfp* pan-neuronal reporter, introduced into ASE-deficient *che-1* mutants, HSN-deficient *sem-4* mutants, or *flp-6/cha-1/tph-1* mutants with defective *flp-6*/acetyl-choline/serotonin signaling, all of which have been implicated in the circuit identified herein. The *Punc-119::gfp* marker is normally expressed in the nerve cells of animals and is apparent from embryo to adult. However, upon GED in tumorous gonads, this marker is also observed in germ cells that acquired neuronal fate within

the gonad (*Levi-Ferber et al., 2015*). Consistent with the phenomenon of GED, we found expression of the neuronal reporter within the tumorous gonads of animals with a perturbed ASE-AIY-HSN-germline but not in animals with an intact circuit (*Figure 6A*). A similar induction of the *Punc-119::gfp* neuronal marker was also observed in cells within the gonads upon artificial activation of *ire-1* in neurons (via *ire-1* overexpression; *Figure 1D*), upon *tfg-1* RNAi-induced ER stress (*Figure 5D*) or upon *flp-6* deficiency (*Figure 2F*). Taken together with the *C. elegans* connectome and epistasis analysis, we conclude that these neurons establish a neuronal circuit whose integrity prevents ectopic differentiation of the germline into somatic cells in the tumorous gonad.

### *flp-6* deficiency suppresses the germline tumor in *gld-1*-deficient animals

We have previously demonstrated that ER stress-induced germ cell transdifferentiation renders apoptosis-resistant cells into apoptosis-sensitive cells, enabling their removal from the tumorous gonad and thus improving the health of the animals (*Levi-Ferber et al., 2015*). Since *flp-6* deficiency also induces germ cell transdifferentiation even in the absence of ER stress, we wondered whether it would similarly improve the health of the animals. Therefore, we measured the effect of *gld-1*-induced germline tumor on the lifespan and mobility of wild-type and *flp-6(-)* animals. We found that in wild-type animals treatment with *gld-1* RNAi shortened the lifespan by approximately 30% (*Figure 6C*). In contrast, treatment with *gld-1* RNAi did not significantly alter lifespan of *flp-6(-)* mutants compared to control RNAi treatment (*Figure 6C*). One of the consequences of the expanding germline tumor is the increased rigidity of the animal's body, which leads to a gradual paralysis. Accordingly, less paralyzed animals were detected among the *flp-6*-defeicient animals compared to wild-type animals upon treatment with *gld-1 RNAi* (*Figure 6D*). Altogether, these results further support the concept that GED can suppress the germline tumor and promote both lifespan and healthspan.

## Discussion

Previous studies demonstrated the tendency of abnormal tumorous gonads to form germline-derived teratoma (*Ciosk et al., 2006*). Here, we demonstrate that cell differentiation decisions in the tumorous germline of animals deficient in the germline-specific translation repressor *gld-1* can be regulated by neuronal cues that are sensitive to ER stress. Interestingly, we demonstrate that the propensity of the tumorous germline to differentiate into somatic cells rather than to maintain pluripotency is actively counteracted by neuronal cues of a newly identified germline differentiation-inhibitory signaling circuit.

The germline differentiation-inhibitory neuronal circuit identified in this study includes the sensory neuron ASE, the interneuron AIY, and the motor neuron HSN, which synapses at the vulva. Furthermore, we identified the central signaling molecules required for the active suppression of GED by this neuronal circuit. While ASE communicates via the neuropeptide FLP-6, AIY communicates via the neurotransmitter Ach and HSN communicates via serotonin. Deficiency in any of these neurons or signaling molecules comprising the germline differentiation-antagonizing circuit resulted in extreme ectopic differentiation of the germline in animals with a germline tumor (see model, *Figure 6B*). Interestingly, additional reproduction-related decisions are determined non-autonomously by neuronal signals. For example, the ASI sensory neuron regulates germline apoptosis, germline mitosis, and sperm activity (*Levi-Ferber et al., 2014*; *Dalfó et al., 2012*; *McKnight et al., 2014*). Nevertheless, in tumorous *gld-1* animals, shifting germline fate from a pluripotent state to an ectopically differentiated somatic state is regulated by a distinct neuronal circuit, governed by the ASE sensory neuron. This implies that separate sets of neurons can provoke divergent responses in the *C. elegans* germline. ASE is a gustatory neuron, known mainly to mediate chemotaxis toward water-soluble cues, including salt ions such as $Na^+$ and $Cl^-$ (*Bargmann and Horvitz, 1991*). While it may be puzzling why a salt-sensing neuron would control germline fate, previous studies demonstrated that the sensing of salt may be coupled to the presence of food (*Tomioka et al., 2006*). In turn, food availability is an important determinant of reproductive physiology.

The most downstream signaling molecule identified in the neuronal circuit that controls germline pluripotency in the tumorous germline is the conserved neurotransmitter serotonin. In *C. elegans*, serotonin causes dramatic behavioral effects including inhibition of locomotion, stimulation of egg

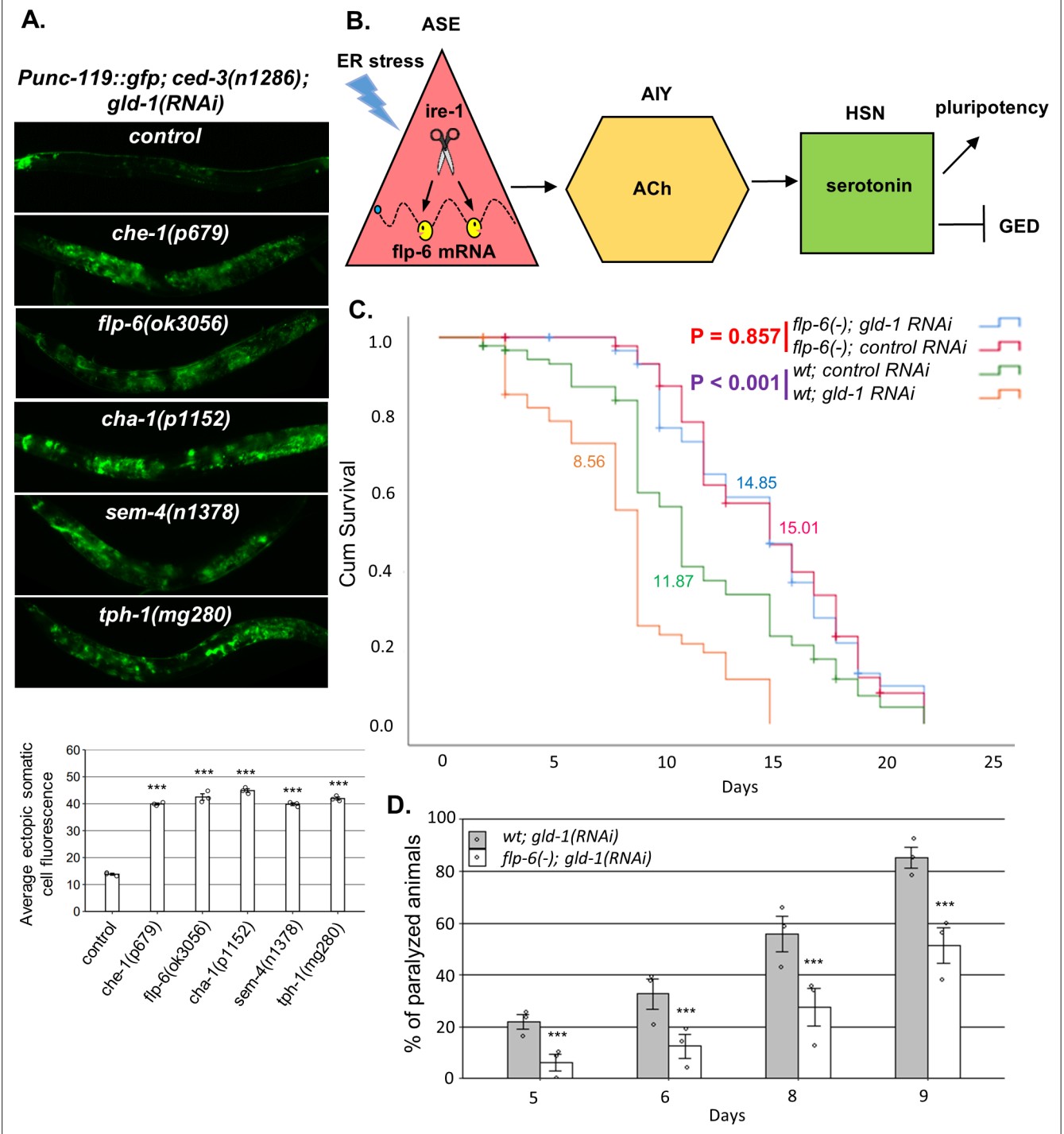

**Figure 6.** A neuronal circuit actively prevents germline ectopic differentiation (GED). (**A**) Abnormal expression of the neuronal reporter *Punc-119::gfp* within the tumorous gonads of *gld-1* RNAi-treated animals with a perturbed ASE-AIY-HSN-germline circuit compared to animals with an intact circuit. *che-1* and *sem-4* mutants are ASE and HSN-deficient, respectively. *flp-6* mutants are deficient in the FLP-6 neuropeptide. *cha-1* and *tph-1* mutants are compromised in acetylcholine (Ach) and serotonin production. Asterisks mark one-way ANOVA followed by Tukey's post hoc analysis of p<0.001 relative to control *ced-3(-)* animals treated with *gld-1* RNAi. Triple asterisks mark significant results resulting in a twofold change or more. (**B**) Summarizing model of ASE-AIY-HSN-germline circuit which actively maintains germline pluripotency and suppresses ectopic germline differentiation in *gld-1* tumorous animals. Implicated neurons and signaling molecules are indicated. Endoplasmic reticulum (ER) stress suppresses this circuit by regulated Ire1-dependent decay (RIDD)-mediated destabilization of the *flp-6* transcript in the ASE neuron. This releases the inhibition from the tumorous germ cells that acquire somatic fate by default. (**C**) Lifespan analysis of wild-type animals and *flp-6* mutants treated with either *gld-1* RNAi to induce tumor formation or with control RNAi. Lifespan shortening was significantly suppressed in *flp-6* mutants treated with *gld-1* RNAi. Mean lifespan is indicated

*Figure 6 continued*

within each graph. Log rank (Mantel–Cox) p<0.001 between wild-type animals treated with either *gld-1* RNAi or with control RNAi is indicated in purple. Log rank (Mantel–Cox) p=0.857 between *flp-6* mutants treated with either *gld-1* RNAi or with control RNAi is indicated in red. See **Supplementary file 2** for statistical data of two additional replicates. (**D**) Paralysis assay in wild-type animals and *flp-6* mutants treated with *gld-1* RNAi. 90 synchronized adult animals per genotype were placed on *gld-1* RNAi plates and their paralysis was scored on days 5, 6, 8, and 9. Bar graphs present percentage of paralyzed animals. At all timepoints, the *flp-6* deficiency resulted in a significant decrease in the paralysis of the tumorous animals in comparison to wild-type animals. Proportions of paralyzed animals were compared between different genotypes in three replicates using the Cochran–Mantel–Haenszel test. Triple asterisks mark significant reduction of at least twofold change in the paralysis relative to wild-type animals treated with *gld-1* RNAi. Note that in the lifespan and paralysis, experiments shown in (**C**) and (**D**) were done in *ced-3(+)* background to allow the apoptosis-mediated clearance of the ER stress-induced aberrant somatic cells in the gonad.

laying, and pharyngeal pumping and food-related behaviors (**Weinshenker et al., 1995**; **Horvitz et al., 1982**; **Ségalat et al., 1995**; **Sawin et al., 2000**; **Rogers et al., 2001**; **Niacaris and Avery, 2003**; **Waggoner et al., 1998**; **Mendel et al., 1995**). While all these effects are modulated by the same molecule, they lead to different independent outcomes. Interestingly, none of the four dedicated *C. elegans* serotonin receptors is required for GED prevention. This may be due to redundancy or due to involvement of other, less specific, serotonin receptors in this circuit. Another layer of specificity may be encoded in the local levels and production site of serotonin. In support of this, serotonin production specifically by the HSN neuron is exclusively critical for preventing GED. Furthermore, migration-defective HSN does not effectively prevent GED, highlighting the importance for the correct location of the synapse of HSN near the vulva. Altogether, this suggests that the site and local concentrations of serotonin release play a key role in GED prevention, which differentiates this function from those of other serotonin signaling events. Furthermore, given the tumor-suppressive effects of GED (**Levi-Ferber et al., 2015**) and consistent with studies that suggest that serotonin levels can play a crucial role in human cancer (**Sarrouilhe and Mesnil, 2019**), this work supports a role for the serotoninergic system in tumor progression, rendering it a potential chemotherapeutic target.

Similarly to GED induction by perturbations in any of the components of the germline differentiation-antagonizing neuronal circuit, ER stress also induces GED within the tumorous gonads, leading to suppression of the germline tumor (**Levi-Ferber et al., 2015**). Our present findings provide mechanistic insights into the underlying mechanism of ER stress-induced GED. Specifically, we show that ER stress-induced GED is not the consequence of ER stress per se within the germ cells, but rather the indirect result of activation of the IRE-1 endoribonuclease in the ASE neuron. We further show that the *flp-6* transcript has the structural requirements for RIDD, undergoes direct 4μ8c-inhibitable and stem-loop sequence-dependent cleavage by purified recombinant IRE1 in vitro, and is destabilized in an *ire-1*-dependent manner in vivo, indicating that the *flp-6* transcript is a target of RIDD. Furthermore, RIDD-mediated destabilization of the *flp-6* transcript is essential for ER stress-induced GED as it cannot be induced in *flp-6* stem-loop mutants, whose *flp-6* transcripts are RIDD-resistant. To the best of our knowledge, this is the first demonstration of a RIDD substrate in *C. elegans*. Furthermore, since the *flp-6* transcript encodes the ASE-produced neuropeptide that controls the GED regulatory circuit, this disrupts the germline differentiation-antagonizing neuronal circuit, which can no longer prevent the acquisition of somatic fate by the germline tumor cells. This model is consistent with our previous findings that it is not the stress itself that regulates germ cell fate, but rather the activation of the ER stress sensor IRE-1, which in turn transduces an *xbp-1*-independent signal to promote GED and limit the progression of the germline tumor (**Levi-Ferber et al., 2015**). Whereas our findings clearly implicate RIDD in ER stress-induced GED, this study has not explored the conceivable implication of additional *xbp-1*-independent signaling pathways regulated by IRE-1 such as the JNK/TRAF2 pathway (**Urano et al., 2000**). Altogether, this is the first evidence that neuronal circuits can be disrupted by RIDD to affect tumor progression by regulating cross-tissue communication.

What could be the benefit in a germline differentiation-inhibitory neuronal circuit? We speculate that under physiological conditions, in a non-tumorous background, such a circuit may be beneficial for preventing precocious differentiation of germline cells prior to fertilization. In the context of animals with a tumorous germline, we have previously demonstrated that acquisition of somatic fate by the tumorous germ cells limits the progression of the tumor by reducing its mitotic capacity and restoring the ability of the cells to execute apoptosis (**Levi-Ferber et al., 2015**). The current identification of an ER stress-regulated neuronal circuit that controls this transition in a distal tumorous tissue

suggests that targeting such neuronal circuits and targeting the RNase activity of IRE1 may be useful as therapeutic interventions for tumor suppression. These findings may be relevant to human tumors as well since many human tumors, originating from a variety of major organs, are innervated, and this innervation is thought to contribute to the pathophysiology of cancer progression (*Hutchings et al., 2020*; *Boilly et al., 2017*).

## Materials and methods

A list of strains and reagents used in this study is provided Appendix 1—key resources table.

### RNA interference

Bacteria expressing dsRNA were cultured overnight in LB containing tetracycline and ampicillin. Bacteria were plated on NGM plates containing 5 mM IPTG and 50 µg/ml carbenicillin. RNAi clone identity was verified by sequencing. Eggs were placed on plates and synchronized at day 0 (L4). The efficacy of the *tfg-1* RNAi was confirmed by the animals' reduced body size (*Witte et al., 2011*) and the induction of the ER stress reporter *Phsp-4::gfp* (*Levi-Ferber et al., 2014*; *Witte et al., 2011*). The efficacy of the *ced-3* RNAi was confirmed by the lack of apoptotic corpses in the gonads. The efficacy of the *gld-1* RNAi was confirmed by the tumorigenicity of the gonads and the absence of oocytes and embryos. For double or triple RNAi mixtures, the relative amount of each RNAi bacteria was kept equal between samples by growing the bacterial cultures overnight and then supplementing the relative amount of control RNAi as needed. Note that we have previously shown that GLD-1 protein levels were efficiently and similarly reduced in all RNAi combinations involving *gld-1* RNAi. including single RNAi treatment, double RNAi treatment. and triple RNAi treatment (see Figure 1—figure supplement 2 in *Levi-Ferber et al., 2015*).

### Fluorescence microscopy and quantification

To follow expression of fluorescent transgenic markers, transgenic animals were anesthetized on 2% agarose pads containing 2 mM levamisol. Images were taken with a CCD digital camera using a Nikon 90i fluorescence microscope. For each trial, exposure time was calibrated to minimize the number of saturated pixels and was kept constant through the experiment. The NIS element software was used to quantify mean fluorescence intensity as measured by intensity of each pixel in the selected area within the gonad.

To determine the fraction of the gonad area occupied by ectopic cells, day 4 animals were fixed and stained with DAPI. The NIS element software was used to manually select and quantify the gonad area as well as the area within the gonad that was occupied by abnormal DAPI-stained nuclei in the animals. Percent of gonad area occupied by large nuclei is the ratio of these two paired measurements.

### Statistical analysis

Error bars represent the standard error of the mean (SEM) of independent biological replicates unless indicated otherwise. For a simple comparison between two data sets, p-values were determined using unpaired Student's t-test, assuming unequal variances. For multiple comparisons, between multiple data sets, groups and/or treatments were compared using one-way or two-way or nested one-way ANOVA followed by a Tukey's post hoc analysis. Normality of residuals assumption was assessed with residuals plots. For lifespan experiments, p-values were calculated using the log rank (Mantel–Cox) analysis. For paralysis assay, p-values were calculated using the Cochran-Mantel–Haenszel test. See *Supplementary file 2* for statistical data. Significance threshold was set as $p < 0.01$ and marked with an asterisk. Significant changes beyond twofold change are marked by three asterisks.

### Quantitative RT-PCR analysis for *flp-6*

Animals were raised at 20 °C until day 1 of adulthood. On day 1, animals were collected for RNA extraction. RNA extraction, purification, and reverse transcription were carried out using standard protocols. Real-time PCR was performed using Maxima SYBR (Fermentas) in a StepOnePlus instrument. Transcript levels of *act-1* were used for normalization. For *act-1* and *flp-6* primers, see Appendix 1—key resources table.

## DAPI staining

Adult worms (at the ages of either day 1, 4, 5, or 10) were collected from agar plates and were washed with M9 solution to remove *Escherichia coli* bacteria. Permeabilization of the animals was performed by freezing them at –80 °C. For fixation, worms were washed with cold methanol and incubated for 15 min at –20 °C. The fixed animals were then washed twice with PBSTx1 and stained with 1 µg/ml 4',6-diamidino-2-phenylindole (DAPI; Sigma) solution for 30 min. Worms were washed two times with PBSTx1 to remove excess staining and observed under the fluorescent microscope to quantify ectopic differentiated nuclei within the animals' gonads.

## Serotonin treatment

Nematodes were grown and assayed at room temperature on standard NGM seeded with *E. coli* strain OP50 as a food source till day 1 of adulthood. For drug experiments, 5-hydroxytryptamine creatinine sulfate complex (Sigma) was added to NGM agar plates to a final concentration of 20 mM. Day 1 animals were grown till day 4 on the 20 mM 5-HT plates and washed for DAPI staining procedure at day 4 of adulthood.

## *flp-6(biu100)* generation

In order to generate the *flp-6(biu100)* mutated strain, we employed a previously described CRISPR/Cas9 genome engineering protocol (*Paix et al., 2015*). Briefly, tracrRNA and two crRNAs, targeting the *dpy-10* and the *flp-6* loci, were mixed with a recombinant cas9 (IDT), ssODN repair template to introduce a dominant mutation into the *dpy-10* locus, and ssODN targeting *flp-6* (see Appendix 1—key resources table for sequences). Rol/dpy F1 progeny were singled and screened by PCR using the primers GTAAGAGCGCTTACATGAGA and TTGAGGTCCAGATCGCTTTC with an annealing temperature of 62 °C, conditions in which only the *flp-6(biu100)* allele is amplified but not the wild-type *flp-6*. Plates with a positive PCR signal were Sanger-sequenced to validate homozygosity of the *flp-6(biu100)* allele. Strains were outcrossed three times to remove the *dpy-10* mutation and reduce other background mutations.

## Histamine treatment

As previously described (*Pokala et al., 2014*), eggs from four histamine-inducible interneurons silencing mutants were grown on non-histamine OP50 seeded plates until L4 stage. At L4 stage, worms were moved to histamine-containing plates (represented by black bars in *Figure 4A* as histamine+) or to non-histamine plates (represented by gray bars in *Figure 4A* as histamine-) until day 4 of adulthood. At day 4 of adulthood, worms were examined for ectopic large nuclei within the gonads using DAPI staining. 1 M histamine (HA) stock was prepared by dissolving 1.85 g histamine dihydrochloride (Sigma-Aldrich H7250) per 10 ml water. Experiments were performed in plates contacting a final concentration of 10 mM histamine.

## Purification of phosphorylated IRE1α KR fractions

Fully phosphorylated human IRE1α KR was produced as follows: human IRE1α KR (G547-L977) was expressed in SF9 cells as an N-terminal His6-tagged fusion protein with a TEV protease cleavage site using an intracellular BEVS expression vector. Cell pellet was resuspended in lysis buffer containing 50 mM HEPES pH 8.0, 300 mM NaCl, 10% glycerol, 1 mM $MgCl_2$, 1:1000 benzonase, EDTA-free PI tablets (Roche), 1 mM TCEP, and 5 mM imidazole. Sample was lysed by sonication, centrifuged at 14,000 rpm for 45 min, and the supernatant filtered through a 0.8 µm Nalgene filter. Cleared supernatant was bound to Ni-NTA Superflow beads (Qiagen) by gravity filtration. Beads were washed in lysis buffer supplemented with 15 mM imidazole, followed by protein elution in lysis buffer containing 300 mM imidazole. The eluate was incubated with TEV protease overnight at 4 °C. The sample of IRE1α KR protein was diluted 1:10 in 50 mM HEPES pH 7.5, 50 mM NaCl, 1 mM TCEP and then loaded onto a 5 ml pre-packed QHP column (GE Healthcare). Isolation of fully phosphorylated IRE1α KR was achieved by eluting the protein with a very shallow gradient (50–300 mM NaCl over 70 CV). Fully phosphorylated fraction was collected separately and confirmed (MW +240) by LC-MS.

## RNA cleavage assay

1 µg of T7 RNA generated was digested at room temperature by 1 µg of human IRE-1α KR recombinant protein (~0.8 µM) for 15 min in RNA cleavage buffer (HEPES pH 7.5 20 mM; potassium acetate 50 mM; magnesium acetate 1 mM; TritonX-100 0.05% [v/v]). The total volume of the reaction is 25 µl. The digestion was then complemented by an equal volume of formamide and heated up at 70 °C for 10 min to linearize the RNA. After linearization, the mixture was immediately placed on ice for 5 min, and then 20 µl was run on 3% agarose gel at 160 V for 50 min. If inhibitor was used, it was incubated with IRE-1α KR for 40 min on ice prior to RNA digestion. Gels were visualized on either a Bio-Rad Molecular Imager ChemiDoc ZRS+. The T7 RNA transcripts was generated from templates containing a wild-type *flp-6* hairpin consensus (atCaGCgtat) or scrambled sequence replacing the original hairpin consensus (ggattacgta) templates of *flp-6*. cDNA was amplified, adding T7 sequence to forward (T7-FLP6-f) and BGH sequence to reverse (BGH-FLP6-r) primers, and subsequently in vitro transcribed using HiScribe T7 Quick High Yield RNA Synthesis Kit from NEB (#E2050S). The IRE-1 RNase-specific inhibitor 4µ8c was used at 5 µM.

| | |
|---|---|
| T7-FLP6-f | TAATACGACTCACTATAGGGCGGCCGGATGAACTCTCGTGGGTTGA |
| BGH-FLP6-r | TAGAAGGCACAGTCGAGGTTATCGTCCGAATCTCATGTATGC |

## Molecular cloning

*Pire-1::ire-1, Prgef-1::ire-1, Pdaf-7::ire-1, Pdaf-28::ire-1* and *Pmyo-3::ire-1* plasmids have been previously described (**Levi-Ferber et al., 2014**). *Psrh-142::flp-6* has been previously described (**Li et al., 2013**).

*Peat-4::ire-1, Punc-25::ire-1, Pche-12::ire-1* and *Pdat-1::ire-1* plasmids have been generated as follows: the *ire-1* coding sequence was amplified from cDNA and cloned into the *Nhe*I and *Kpn*I sites in the L3691 plasmid. Genomic DNA-amplified promoters were inserted upstream of the *ire-1* coding sequence (CDS) as follows: *Peat-4* (~3 kb), *Punc-25* (~1.6 kb), *Pche-12* (~ 1 kb), and *Pdat-1* (~0.7 kb) were inserted in the SphI/NotI restriction sites. *Pges-1* (~3 kb) was inserted in the SphI/KpnI restriction sites.

To generate *Pche-1::flp-6,* the *flp-6* coding sequence was amplified from cDNA and cloned into the XmaI and KpnI sites in the Andy-Fire L3691 vector. *Pche-1* (ASE specific, ~1.4 kb) promoter was cloned into NotI and XmaI sites upstream to *flp-6* cDNA, replacing the original *mec-7* promoter.

To generate *Pttx-3::cha-1* and *Pttx-3::unc-17*, the *cha-1* coding sequence and the *unc-17* coding sequence were amplified from cDNA and cloned into the *Kpn*I/*Sph*I sites downstream of the *Pttx-3*[promB] regulatory element in the *Pttx-3::ire-1* vector, previously described (**Levi-Ferber et al., 2014**), replacing the *ire-1* coding sequence.

Germline transformations were performed by injection of 25 ng/ml plasmids and 100 ng/ml of *rol-6(su1006)* as a co-transformation marker.

## Acknowledgements

Some nematode strains were provided by the Caenorhabditis Genetics Center, which is funded by the NIH National Center for Research Resources and by Dr. Shohei Mitani, National Bioresource Project for the nematode, Tokyo Women's Medical University School of Medicine, Japan. We thank Prof. Cori Bargmann (Rockefeller University, USA) for the HisCl-related strains as well as for the Cre-Lox-specific knockouts of *tph-1*. We thank Prof. Kang Shen (Stanford University, USA) for IRE-1 structure function strains (*wy782, wy762*). We thank Prof. Hannes E Bulow (Albert Einstein College of Medicine, USA) for helpful discussions. We thank Prof. Dayong Wang (Southeast University, China) for the *flp-6* rescue in ADF construct *(Psrh-142::flp-6).* We thank Prof. Chris Li (The City College of New York, USA) for *flp-4(yn35)* mutant strain. This work was supported by the Israel Science Foundation (grant number 1571/15 to SHK). The funders had no role in study design, data collection and analysis, decision to publish, or preparation of the manuscript.

# Additional information

### Competing interests
Adrien Le-Thomas: Adrien Le-Thomas is affiliated with Genentech. The author has no other competing interests to declare.. Avi Ashkenazi: Avi Ashkenazi is affiliated with Genentech. The author has no other competing interests to declare.. The other authors declare that no competing interests exist.

### Funding

| Funder | Grant reference number | Author |
|---|---|---|
| Israel Science Foundation | 1571/15 | Sivan Henis-Korenblit |

The funders had no role in study design, data collection and interpretation, or the decision to submit the work for publication.

### Author contributions
Mor Levi-Ferber, Conceptualization, Formal analysis, Investigation, Methodology, Visualization, Writing - original draft; Rewayd Shalash, Yehuda Salzberg, Maor Shurgi, Investigation; Adrien Le-Thomas, Investigation, Visualization; Jennifer IC Benichou, statistical analysis and data presentation; Avi Ashkenazi, Investigation, Methodology, Writing – review and editing; Sivan Henis-Korenblit, Conceptualization, Formal analysis, Funding acquisition, Investigation, Methodology, Project administration, Supervision, Writing - original draft

### Author ORCIDs
Avi Ashkenazi (iD) http://orcid.org/0000-0002-6890-4589
Sivan Henis-Korenblit (iD) http://orcid.org/0000-0001-8023-6336

### Decision letter and Author response
Decision letter https://doi.org/10.7554/eLife.65644.sa1
Author response https://doi.org/10.7554/eLife.65644.sa2

# Additional files

### Supplementary files
• Supplementary file 1. List of sensory neurons included under the *che-1* promoter.
• Supplementary file 2. Statistical data.
• Transparent reporting form

### Data availability
All data generated or analysed during this study are included in the manuscript.

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

# Appendix 1

## Appendix 1—key resources table

| Reagent type (species) or resource | Designation | Source or reference | Identifiers | Additional information |
|---|---|---|---|---|
| Strain, strain background (*Caenorhabditiselegans*) | N2 | Caenorhabditis Genetics Center | Wild type | |
| Strain, strain background (*C. elegans*) | SHK124 | This paper | *rrf-1(pk1417) I* | x4 outcrosses, strain created in S. Henis-Korenblit lab |
| Strain, strain background (*C. elegans*) | CF2473 | Cynthia Kenyon lab | *ire-1(ok799) II* | x3 outcrosses, strain created in C. Kenyon lab |
| Strain, strain background (*C. elegans*) | SHK27 | *Levi-Ferber et al., 2015* PMID:25340700 | *ire-1(ok799) II; biuEx2[pRF4(rol-6(su1006)); Pire-1::ire-1(cDNA)]; svIs69[Pdaf-28::daf-28::gfp]* | Strain created in S. Henis-Korenblit lab |
| Strain, strain background (*C. elegans*) | SHK185 | *Levi-Ferber et al., 2014* PMID:25340700 | *ire-1(ok799) II; biuEx49[pRF4(rol-6(su1006)); Prgef-1::ire-1(cDNA)]* | Strain created in S. Henis-Korenblit lab |
| Strain, strain background (*C. elegans*) | SHK14 | *Levi-Ferber et al., 2014* PMID:25340700 | *ire-1(ok799) II; biuEx5[pRF4(rol-6(su1006)); Pmyo-3::ire-1(cDNA)]* | Strain created in S. Henis-Korenblit lab |
| Strain, strain background (*C. elegans*) | SHK703 | This paper | *ire-1(ok799) II wdIs52 [F49H12.4::GFP + unc-119(+)]; biuEx50[pRF4(rol-6(su1006)); Pges-1::ire-1(cDNA)]* | Strain created in S. Henis-Korenblit lab |
| Strain, strain background (*C. elegans*) | SHK4 | *Levi-Ferber et al., 2014* PMID:25340700 | *biuEx2[pRF4(rol-6(su1006)); Pire-1::ire-1(cDNA)]* | Strain created in S. Henis-Korenblit lab |
| Strain, strain background (*C. elegans*) | SHK182 | *Levi-Ferber et al., 2014* PMID:25340700 | *biuEx49[pRF4(rol-6(su1006)); Prgef-1::ire-1(cDNA)]* | Strain created in S. Henis-Korenblit lab |
| Strain, strain background (*C. elegans*) | SHK8 | *Levi-Ferber et al., 2014* PMID:25340700 | *biuEx5[pRF4(rol-6(su1006)); Pmyo-3::ire-1(cDNA)]* | Strain created in S. Henis-Korenblit lab |
| Strain, strain background (*C. elegans*) | SHK282 | This paper | *biuEx50[pRF4(rol-6(su1006)); Pges-1::ire-1(cDNA)]* | Strain created in S. Henis-Korenblit lab |
| Strain, strain background (*C. elegans*) | SHK268 | This paper | *ire-1(ok799) II; biuEx11[Pgrd-10::gfp; Pche-12::ire-1(cDNA)]* | Strain created in S. Henis-Korenblit lab |
| Strain, strain background (*C. elegans*) | SHK271 | This paper | *ire-1(ok799) II; biuEx14[Pgrd-10::gfp; Peat-4::ire-1(cDNA)]* | Strain created in S. Henis-Korenblit lab |
| Strain, strain background (*C. elegans*) | SHK270 | This paper | *ire-1(ok799) II; biuEx13[Pgrd-10::gfp; Pdat-1::ire-1(cDNA)]* | Strain created in S. Henis-Korenblit lab |
| Strain, strain background (*C. elegans*) | SHK266 | This paper | *ire-1(ok799) II; biuEx10[Pgrd-10::gfp; Punc-25::ire-1(cDNA)]* | Strain created in S. Henis-Korenblit lab |
| Strain, strain background (*C. elegans*) | SHK15 | *Levi-Ferber et al., 2014* PMID:25340700 | *ire-1(ok799) II; biuEx4[pRF4(rol-6(su1006)); Pdaf-28::ire-1(cDNA)]* | Strain created in S. Henis-Korenblit lab |
| Strain, strain background (*C. elegans*) | SHK256 | This paper | *ire-1(ok799) II; biuEx51[pRF4(rol-6(su1006)); Pdaf-7::gfp; Pdaf-7::ire-1(cDNA)]* | Strain created in S. Henis-Korenblit lab |
| Strain, strain background (*C. elegans*) | SHK592 | This paper | *ire-1(ok799) II; biuEx57[pRF4(rol-6(su1006)); Pche-1::ire-1(cDNA)]* | Strain created in S. Henis-Korenblit lab |
| Strain, strain background (*C. elegans*) | SHK593 | This paper | *ire-1(ok799) II; biuEx58[pRF4(rol-6(su1006)); Pops-1::ire-1(cDNA)]* | Strain created in S. Henis-Korenblit lab |
| Strain, strain background (*C. elegans*) | OH10434 | Caenorhabditis Genetics Center | *otIs232 [che-1p::mcherry(C. elegans-optimized)::che-1 3'UTR + rol-6(su1006)]* | |

*Appendix 1 Continued on next page*

*Appendix 1 Continued*

| Reagent type (species) or resource | Designation | Source or reference | Identifiers | Additional information |
|---|---|---|---|---|
| Strain, strain background (*C. elegans*) | DA572 | Caenorhabditis Genetics Center | *eat-4(ad572)* | |
| Strain, strain background (*C. elegans*) | MT6308 | Caenorhabditis Genetics Center | *eat-4(ky5)* | |
| Strain, strain background (*C. elegans*) | NY119 | Chris Li lab | *flp-4(yn35) II* | |
| Strain, strain background (*C. elegans*) | VC2324 | Caenorhabditis Genetics Center | *flp-6(ok3056) V* | |
| Strain, strain background (*C. elegans*) | FX02427 | Dr. Shohei Mitani, National Bioresource Project for the nematode, Japan | *flp-13(tm2427) IV* | |
| Strain, strain background (*C. elegans*) | PT505 | Caenorhabditis Genetics Center | *flp-20(pk1596) X* | |
| Strain, strain background (*C. elegans*) | VC1982 | Caenorhabditis Genetics Center | *flp-25(gk1016) III* | |
| Strain, strain background (*C. elegans*) | FX03023 | Dr. Shohei Mitani, National Bioresource Project for the nematode, Japan | *nlp-3(tm3023) X* | |
| Strain, strain background (*C. elegans*) | RB1609 | Caenorhabditis Genetics Center | *nlp-5(ok1981) II* | |
| Strain, strain background (*C. elegans*) | FX1880 | Dr. Shohei Mitani, National Bioresource Project for the nematode, Japan | *nlp-14(tm1880) X* | |
| Strain, strain background (*C. elegans*) | FX1888 | Caenorhabditis Genetics Center | *ins-1(tm1888) IV* | |
| Strain, strain background (*C. elegans*) | RB2594 | Caenorhabditis Genetics Center | *ins-22(ok3616) III* | |
| Strain, strain background (*C. elegans*) | FX14756 | Dr. Shohei Mitani, National Bioresource Project for the nematode, Japan | *ins-26(tm1983) I* | |
| Strain, strain background (*C. elegans*) | FX06109 | Caenorhabditis Genetics Center | *ins-32(tm6109) II* | |
| Strain, strain background (*C. elegans*) | RB982 | Caenorhabditis Genetics Center | *flp-21(ok889) V* | |
| Strain, strain background (*C. elegans*) | RB1340 | Caenorhabditis Genetics Center | *nlp-1(ok1469) X* | |
| Strain, strain background (*C. elegans*) | FX02984 | Dr. Shohei Mitani, National Bioresource Project for the nematode, Japan | *nlp-7(tm2984) X* | |
| Strain, strain background (*C. elegans*) | VC1309 | Caenorhabditis Genetics Center | *nlp-8(ok1799) I* | |
| Strain, strain background (*C. elegans*) | FX06232 | Dr. Shohei Mitani, National Bioresource Project for the nematode, Japan | *nlp-10(tm6232) III* | |

*Appendix 1 Continued on next page*

*Appendix 1 Continued*

| Reagent type (species) or resource | Designation | Source or reference | Identifiers | Additional information |
|---|---|---|---|---|
| Strain, strain background (*C. elegans*) | VC1063 | Caenorhabditis Genetics Center | *nlp-15(ok1512) I* | |
| Strain, strain background (*C. elegans*) | SHK497 | This paper | *biuEx52[pRF4(rol-6(su1006)); Pche-1::flp-6(cDNA)]; flp-6(ok3056) V* | Strain created in S. Henis-Korenblit lab |
| Strain, strain background (*C. elegans*) | SHK498 | This paper | *biuEx53[pRF4(rol-6(su1006)); Psrh-142::flp-6]; flp-6(ok3056) V* | Strain created in S. Henis-Korenblit lab |
| Strain, strain background (*C. elegans*) | PR672 | Caenorhabditis Genetics Center | *che-1(p672) I* | |
| Strain, strain background (*C. elegans*) | PR674 | Caenorhabditis Genetics Center | *che-1(p674) I* | |
| Strain, strain background (*C. elegans*) | MT633 | Caenorhabditis Genetics Center | *lin-11(n389) I; him-5(e1467)V* | |
| Strain, strain background (*C. elegans*) | MT1196 | Caenorhabditis Genetics Center | *lin-11(n566) I* | |
| Strain, strain background (*C. elegans*) | CF2260 | Cynthia Kenyon lab | *Zcls4[Phsp-4::gfp] V* | |
| Strain, strain background (*C. elegans*) | SHK314 | This paper | *Zcls4[Phsp-4::gfp] V; flp-6(ok3056) V* | Strain created in S. Henis-Korenblit lab |
| Strain, strain background (*C. elegans*) | SHK315 | This paper | *ire-1(ok799) II, flp-6(ok3056) V* | Strain created in S. Henis-Korenblit lab |
| Strain, strain background (*C. elegans*) | NY2067 | Caenorhabditis Genetics Center | *ynIs67[Pflp-6::gfp] III; him-5(e1490) V* | |
| Strain, strain background (*C. elegans*) | SHK403 | This paper | *ynIs67[Pflp-6::gfp] III; ire-1(ok799) II* | Strain created in S. Henis-Korenblit lab |
| Strain, strain background (*C. elegans*) | SHK474 | This paper | *biuEx54[pRF4(rol-6(su1006)); Pche-1::flp-6(cDNA)::gfp]* | Strain created in S. Henis-Korenblit lab |
| Strain, strain background (*C. elegans*) | SHK491 | This paper | *biuEx52[pRF4(rol-6(su1006)); Pche-1::flp-6(cDNA)]; ire-1(ok799) II* | Strain created in S. Henis-Korenblit lab |
| Strain, strain background (*C. elegans*) | TV13656 | Kang Shen lab | *ire-1(wy782) II* | |
| Strain, strain background (*C. elegans*) | TV13763 | Kang Shen lab | *ire-1(wy762) II* | |
| Strain, strain background (*C. elegans*) | NY7 | Caenorhabditis Genetics Center | *flp-1(yn2) IV* | |
| Strain, strain background (*C. elegans*) | NY16 | Caenorhabditis Genetics Center | *flp-1(yn4) IV* | |
| Strain, strain background (*C. elegans*) | AX1410 | Caenorhabditis Genetics Center | *flp-8(db99) X* | |
| Strain, strain background (*C. elegans*) | VC2016 | Caenorhabditis Genetics Center | *flp-18(gk3063) X* | |
| Strain, strain background (*C. elegans*) | PR1152 | Caenorhabditis Genetics Center | *cha-1(p1152) IV* | |
| Strain, strain background (*C. elegans*) | SHK605 | This paper | *biuEx55[pRF4(rol-6(su1006)); Pttx-3::cha-1]; cha-1(p1152) IV* | Strain created in S. Henis-Korenblit lab |
| Strain, strain background (*C. elegans*) | GG201 | Caenorhabditis Genetics Center | *ace-2(g72) I; ace-1(p1000) X* | |
| Strain, strain background (*C. elegans*) | PR1300 | Caenorhabditis Genetics Center | *ace-3(dc2) II* | |
| Strain, strain background (*C. elegans*) | CB113 | Caenorhabditis Genetics Center | *unc-17(e113) IV* | |

*Appendix 1 Continued on next page*

*Appendix 1 Continued*

| Reagent type (species) or resource | Designation | Source or reference | Identifiers | Additional information |
|---|---|---|---|---|
| Strain, strain background (*C. elegans*) | SHK586 | This paper | *biuEx56[pRF4(rol-6(su1006)); Pttx-3::unc-17]; unc-17(e113) IV* | Strain created in S. Henis-Korenblit lab |
| Strain, strain background (*C. elegans*) | MT9668 | Caenorhabditis Genetics Center | *mod-1(ok103) V* | |
| Strain, strain background (*C. elegans*) | DA1814 | Caenorhabditis Genetics Center | *ser-1(ok345) X* | |
| Strain, strain background (*C. elegans*) | AQ866 | Caenorhabditis Genetics Center | *ser-4 (ok512) III* | |
| Strain, strain background (*C. elegans*) | RB1585 | Caenorhabditis Genetics Center | *ser-7(ok1944) X* | |
| Strain, strain background (*C. elegans*) | MT1446 | Caenorhabditis Genetics Center | *her-1(n695) V* | |
| Strain, strain background (*C. elegans*) | MT5825 | Caenorhabditis Genetics Center | *sem-4(n1378) I* | |
| Strain, strain background (*C. elegans*) | MT3969 | Caenorhabditis Genetics Center | *mig-1(n1652) I* | |
| Strain, strain background (*C. elegans*) | NF69 | Caenorhabditis Genetics Center | *mig-19(k142) II* | |
| Strain, strain background (*C. elegans*) | NF78 | Caenorhabditis Genetics Center | *mig-20(k148) X* | |
| Strain, strain background (*C. elegans*) | SHK492 | This paper | *flp-6(ok3056) V; sem-4(n1378) I* | Strain created in S. Henis-Korenblit lab |
| Strain, strain background (*C. elegans*) | SHK490 | This paper | *flp-6(ok3056) V; her-1(n695) V* | Strain created in S. Henis-Korenblit lab |
| Strain, strain background (*C. elegans*) | FX30280 | Dr. Shohei Mitani, National Bioresource Project for the nematode, Japan | *flp-5(tm10075) X* | |
| Strain, strain background (*C. elegans*) | LC33 | Dr. Shohei Mitani, National Bioresource Project for the nematode, Japan | *bas-1(tm351) III* | |
| Strain, strain background (*C. elegans*) | CX13228 | Cori Bargman Lab | *tph-1(mg280) II; kySi56[ tph-1 genomic rescue] IV* | Strain created in C. Bargman Lab |
| Strain, strain background (*C. elegans*) | CX13576 | Cori Bargman Lab | *tph-1(mg280) II; kySi56[ tph-1 genomic rescue] IV; kyEx4107[egl6::nCre]* | Strain created in C. Bargman Lab |
| Strain, strain background (*C. elegans*) | CX13571 | Cori Bargman Lab | *tph-1(mg280) II; kySi56[ tph-1 genomic rescue] IV; kyEx4077[srh142::nCre]* | Strain created in C. Bargman Lab |
| Strain, strain background (*C. elegans*) | CX13574 | Cori Bargman Lab | *tph-1(mg280) II; kySi56[ tph-1 genomic rescue] IV; kyEx4081[ops1::nCre]* | Strain created in C. Bargman Lab |
| Strain, strain background (*C. elegans*) | CX14909 | Cori Bargman Lab | *kyEx4925 [ttx-3::hisCl1*::sl2::GFP; myo-3::mCherry]* | Strain created in C. Bargman Lab |
| Strain, strain background (*C. elegans*) | CX14849 | Cori Bargman Lab | *kyEx4867 [ins-1::HisCl1::sl2mCherry; unc-122::GFP]* | Strain created in C. Bargman Lab |
| Strain, strain background (*C. elegans*) | CX14908 | Cori Bargman Lab | *kyEx4924 [inx1::hisCl1*::sl2::GFP; myo-3::mCherry]* | Strain created in C. Bargman Lab |

*Appendix 1 Continued on next page*

*Appendix 1 Continued*

| Reagent type (species) or resource | Designation | Source or reference | Identifiers | Additional information |
|---|---|---|---|---|
| Strain, strain background (*C. elegans*) | CX16069 | Cori Bargman Lab | *kyEx5493 [pNP443 (glr3::HisCl1::SL2::mCherry); elt-2:mCherry]* | Strain created in C. Bargman Lab |
| Strain, strain background (*C. elegans*) | DP132 | Caenorhabditis Genetics Center | *edIs6 (punc-119::GFP) IV* | |
| Strain, strain background (*C. elegans*) | SHK659 | This paper | *che-1(p679) I; edIs6 (punc-119::GFP) IV* | Strain created in S. Henis-Korenblit lab |
| Strain, strain background (*C. elegans*) | SHK361 | This paper | *edIs6 (punc-119::GFP) IV; flp-6(ok3056) V* | Strain created in S. Henis-Korenblit lab |
| Strain, strain background (*C. elegans*) | SHK660 | This paper | *cha-1(p1152) edIs6 (punc-119::GFP) IV* | Strain created in S. Henis-Korenblit lab |
| Strain, strain background (*C. elegans*) | SHK661 | This paper | *sem-4(n1378) I; edIs6 (punc-119::GFP) IV* | Strain created in S. Henis-Korenblit lab |
| Strain, strain background (*C. elegans*) | SHK662 | This paper | *tph-1(mg280) II; edIs6 (punc-119::GFP) IV* | Strain created in S. Henis-Korenblit lab |
| Strain, strain background (*C. elegans*) | SHK663 | This paper | *Zcls4[Phsp-4::gfp] V; otIs232 [che-1p::mcherry(C. elegans-optimized)::che-1 3'UTR + rol-6(su1006)]* | |
| Strain, strain background (*C. elegans*) | CF3208 | Cynthia Kenyon lab | *xbp-1(tm2457) III* | X4 outcrosses |
| Strain, strain background (*C. elegans*) | SHK62 | *Levi-Ferber et al., 2014* PMID:25340700 | *xbp-1(tm2457) III; ire-1(ok799) II* | Strain created in S. Henis-Korenblit lab |
| Strain, strain background (*C. elegans*) | SHK152 | *Levi-Ferber et al., 2015* PMID:26192965 | *edIs6 (punc-119::GFP) IV; ced-3(n1286) IV* | Strain created in S. Henis-Korenblit lab |
| Strain, strain background (*C. elegans*) | SHK86 | *Levi-Ferber et al., 2015* PMID:26192965 | *Pmyo-3::GFP; ced-3(n1286)* | Strain created in S. Henis-Korenblit lab |
| Strain, strain background (*C. elegans*) | SHK697 | This paper | *flp-6(biu100)* | Strain created in S. Henis-Korenblit lab |
| Strain, strain background (*C. elegans*) | SHK698 | This paper | *ire-1(ok799) II; otIs232 [che-1p::mcherry(C. elegans-optimized)::che-1 3'UTR + rol-6(su1006)]* | Strain created in S. Henis-Korenblit lab |
| Strain, strain background (*C. elegans*) | SHK699 | This paper | *edIs6 (punc-119::GFP) IV; ced-3(n1286) IV; biuEx49[pRF4(rol-6(su1006)); Prgef-1::ire-1(cDNA)]* | Strain created in S. Henis-Korenblit lab |
| Strain, strain background (*C. elegans*) | SHK584 | This paper | *biuEx57[pRF4(rol-6(su1006)); Pche-1::ire-1(cDNA)]* | Strain created in S. Henis-Korenblit lab |
| Sequence-based reagent | *act-1* Fw | *Cohen-Berkman et al., 2020* PMID:32213289 | qPCR primer | CCAATCCAAGAGAGGT ATCCTTAC |
| Sequence-based reagent | *act-1* Bw | *Cohen-Berkman et al., 2020* PMID:32213289 | qPCR primer | CATTGTAGAAGGTGTG ATGCCAG |
| Sequence-based reagent | *flp-6* Fw | This paper | qPCR primer | GTGAAGTGGAGAGAGA AATGATGA |
| Sequence-based reagent | *flp-6* Bw | This paper | qPCR primer | CCGCTACTTCTCTTTC CAAAACG |
| Chemical compound, drug | TRIzol | Ambion | 15596026 | |

*Appendix 1 Continued on next page*

*Appendix 1 Continued*

| Reagent type (species) or resource | Designation | Source or reference | Identifiers | Additional information |
|---|---|---|---|---|
| Chemical compound, drug | Maxima SYBR GREEN | Thermo Scientific | K0221 | |
| Chemical compound, drug | IPTG | Gold Bio | I2481C | |
| Chemical compound, drug | DAPI | Sigma | D9542 | |
| Chemical compound, drug | Serotonine creatinine sulfate monohydrate | Sigma | H7752 | |
| Chemical compound, drug | Histamine dihydrochloride | Sigma | H7250 | |
| Chemical compound, drug | Levamisol hydrochloride | Sigma | 31742 | |
| Chemical compound, drug | Tunicamycin | Cayman | 11445 | |
| Other | CFX-96 real time system | Bio-Rad | | |
| Software, algorithm | SPSS | SPSS | | |
| Software, algorithm | R statistical environment | R Foundation for Statistical Computing, Vienna, Austria. https://www.R-project.org/. | RRID:SCR_001905 | |
| Sequence-based reagent | tracrRNA | IDT | | |
| Sequence-based reagent | cas9 enzyme | IDT | | |
| Sequence-based reagent | dpy-10 crRNA | IDT | GCTACCATAGGCACCACGAG | |
| Sequence-based reagent | flp-6 crRNA | IDT | AAATCAGCGTATATGCGTTT | |
| Sequence-based reagent | dpy-10 ssODN | IDT | CACTTGAACTTCAATA CGGCAA GATGAGAATGACTGGA AACCGT ACCGCATGCGGTGCCT ATGGTA GCGGAGCTTCACATGG CTTCA GACCAACAGCCTAT | |
| Sequence-based reagent | flp-6 (biu100) ssODN | IDT | tcaaaaatatgttttgcagAAATGAT GAAGCGTAAgagcGCtTAcAT GaGaTTCGGACGTTCTGAC GGTGGAAACCCAATGGAA ATGGAAA | |

