## [Decision Letter]

**Acceptance summary:**

This paper describes how the ER stress sensor Ire1 modulates the ability of *C. elegans* to maintain pluripotency in the germline. Surprisingly, Ire1 regulates ectopic differentiation of germline cells through the degradation of an mRNA substrate in the nervous system. The authors describe both the regulation of this novel substrate of Ire1 and the neural circuit that links Ire1 activity to gonad pluripotency.

**Decision letter after peer review:**

Thank you for submitting your article "Neuronal regulated Ire1-dependent mRNA decay controls germline differentiation in *C. elegans*" for consideration by *eLife*. Your article has been reviewed by 3 peer reviewers, one of whom is a member of our Board of Reviewing Editors, and the evaluation has been overseen David Ron as the Senior Editor. The reviewers have opted to remain anonymous.

Essential Revisions:

1. As pointed out by reviewer 3 below, is unclear whether all of the effects resulting in abnormal nuclei (as visualized by DAPI staining) are a result of ectopic differentiation. The authors should repeat the key experiments underlying the main findings regarding neuronal control of GED, and examine differentiation markers in the gonad.

2. The reviewers agreed that more evidence is needed to demonstrate that flp6 is a target of RIDD. The suggestions for experiments to support this conclusion are as follows: (a) measure flp6 RNA levels upon inducing ER stress, and when Ire1 is expressed in ASE, with the prediction that flp6 RNA levels should decrease; (b) if possible (given potential difficulties with the Xbp1 mutants), demonstrate that flp6 RNA degradation is independent of Xbp; (c) indicate whether the in vitro cleavage assay results in cleavage at the putative cleavage site described by the authors, and test whether mutation of this site prevents degradation of the flip6 RNA in vitro and in vivo.

3. The authors should test the specificity of serotonin for the ASE-AIY-HSN-germline circuit. Can the effects of tgf-1 RNAi on GED also be blocked by serotonin? Does serotonin interfere with the mex-3 RNAi GED phenotype?

4. Given the potential therapeutic aspect of this work and the previous findings that Ire1 activation allows animals with tumors to live longer, the authors should test how the flp6 mutant affects survival and/or paralysis in gld-1-RNAi treated animals.

*Reviewer #1 (Recommendations for the authors):*

My main suggestion for an aspect of this paper that could be strengthened is included in my public review. To address this, I would suggest that the authors determine whether flp6 degraded when Ire1 is specifically expressed in the ASE as in figure 3H.

A second suggestion is about the in vitro assay. The authors also show an experiment where Ire1 appears to cleave the FLP6 RNA in Figure 3D. To further support that this is a specific cleavage event, it would be nice to know if the sizes of the cleavage products in Figure 3D are consistent with cleavage at the stem loop structure described by the authors. Even more conclusive would be if the authors showed that mutating the critical nucleotide(s) in the cleavage sites prevented cleavage, or that a second RNA without such a site is not cleaved in this assay.

*Reviewer #2 (Recommendations for the authors):*

1. A central point of the article is to show for the first time a RIDD target in *C. elegans*. Although their findings are consistent with this conclusion, more evidence is required to firmly establish flp-6 as a RIDD target. Measuring the levels of flp-6 in other ER stress conditions, such as tunicamycin treatment, would help to build a convincing case that flp-6 is a genuine RIDD target. The authors establish that flp-6 can act as a direct target of the IRE-1 endoribonuclease in vitro, but not that it is actually a direct target in vivo; mutating the predicted stem loop structure and demonstrating an inhibition of flp-6 loss and downstream effects on GED would solidify the authors argument. It is also important to exclude any potential role of the XBP-1-mediated UPR in flp-6 loss, by measuring flp-6 levels in xbp-1(-) mutants – although this might be a difficult result to interpret as previous author data indicates that xbp-1(-) mutants already display a basal increase in the ER stress levels (Safra M et al., 2013; Levi-Ferber M et al., 2015). Also, in addition to RIDD and xbp-1 splicing, IRE-1 has other functions in the cell, such as regulating *JNK*/*TRAF2* signaling. Authors should exclude (or at least discuss) any potential role of *JNK*/*TRAF2* in the regulation of GED by IRE-1.

2. The authors did a fantastic job in mapping the neuronal circuits controlling GED. Testing whether the effects of tgf-1 RNAi on GED can also be blocked by serotonin would help to support the model proposed in Figure 6. In addition, checking whether serotonin interferes with the mex-3 RNAi GED phenotype would help demonstrate serotonin specificity for the ASE-AIY-HSN-germline circuit.

3. The authors emphasize in the discussion the potential therapeutic aspect of the work. In their previous paper, they were able to show that IRE-1 activation allows animals with tumors to live longer (Levi-Ferber et al., 2015). To further support the therapeutic aspect of this work, I suggest that the authors test the effects of flp-6 mutant in survival and/or paralysis in gld-1-RNAi treated animals.

*Reviewer #3 (Recommendations for the authors):*

Figure 1: The rrf-1(pk1417) I used, according to the publications cited by the authors, can undergo at least partial RNAi mediated knockdown in the intestine, and show elevated expression of, at least some, transgenic arrays. In addition, other artifacts of triple RNAi have not been excluded in the studies shown here.

The images are low resolution, and the germline looks largely disturbed and DAPI is used as a proxy for GED. While it is possible some of the cells are differentiating ectopically (vis-a-vie the authors' previous publication), that needs to be shown.

Authors should not use 'abnormal nuclei' interchangeably with GED (eg. nuclei in tph-1 mutants in Figure 5)

IRE-1 is supposed to be required for GED. However, Figure 1B, tgf-1 (RNAi); ire-1 (-) animals appear to have GED.

The effects of IRE-1 manipulation on ER stress needs to be carefully assayed using qRT-PCR, XBP-1 splicing etc., before one can conclude, as the authors do that " active IRE-1 signaling rather than ER stress" modulates GED. In this regard, it would be important for the authors to spell out what they mean by ER stress. Where? Which sensors etc.

Controls for o/e of ire-1 in each of the tissues alone should be shown.

Figure 2: Previous studies by other groups have shown that overexpressing spliced XBP-1 in neurons of *C. elegans* induced ER stress in other tissues (shown in intestine) through neurotransmitter release (unc-13 mechanism). Even though the authors believe that the phenomenon they are studying is XBP-1 independent, and dependent on IRE-1 RIDD activity, the use a mutant IRE-1 unable to splice XBP-1 would be the conclusive way to demonstrate this. In its absence, the observations could be due to IRE-1 induced XBP-1 splicing.

Figure 3: It is unclear why the ire-1(-) animals not show the same phenotype as flp-6 (-) animals? Without flp-6, GED is constitutive (in a gld-1, ced-3 background). Without ire-1 GED is inhibited (in same background). Yet in a flp-6; ire-1(-) strain GED is activated similar to flp-6 null-the reason for this is not clear.

How do authors reconcile ire-1 independence of GED in flp-6 mutants with an ire-1 dependent hsp-4::GFP is a transgene that is expressed mainly in intestine, and as such is an incomplete readout of ER stress. Therefore, its sole use to establish that flp-6(-) animals do not activate ER stress is not supported.

Neuronal expression of the Zcls4s has not been previously established. Images of expression that was quantified in ASE should therefore be shown.

Is the FLP-6 translational reporter expression levels restored in the wy762 mutant?

IRE-1 RIDD activity has not been demonstrated in *C. elegans*, and if FLP-6 is indeed an IRE-1 RIDD target, this would be exciting new advance in the field. Therefore these data while provocative, should be validated in independent ways. A convincing method to show IRE-1 RIDD activity would be to re-express an IRE-1 kinase domain only (or mutant without nuclease activity) to see if does not decrease expression of the flp-6 translational reporter. For instance, does is flp-6 lacking the stem loop also a RIDD substrate in the in vitro assays? What is the efficacy of flp-6 processing the RIDD reaction ? How well would known RIDD substrates be cleaved in this assay? If ER-UPR is activated in the FLP-6 translational reporter could be downregulated and explain the results.

---

## [Author Response]

Essential Revisions:1. As pointed out by reviewer 3 below, is unclear whether all of the effects resulting in abnormal nuclei (as visualized by DAPI staining) are a result of ectopic differentiation. The authors should repeat the key experiments underlying the main findings regarding neuronal control of GED, and examine differentiation markers in the gonad.

To further demonstrate that the conditions resulting in accumulation of abnormal nuclei (as visualized by DAPI staining) in the gonads are a result of ectopic differentiation we added experiments demonstrating expression of somatic markers within the gonad, under similar conditions. Expression of a neuronal marker within the gonads of animals over-expressing ire-1 in their neurons is shown in Figure 1D. Expression of a neuronal and a muscle marker within the gonads of treated with flp-6 RNAi is shown in Figure 2F. Expression of a neuronal marker within the gonads of treated with tfg-1 RNAi is shown in Figure 5D, as well as its suppression by serotonin supplementation. Expression of a neuronal marker within the gonads of animals in which the ASE-AIY-HSN neuronal circuit has been impaired by altering the identity of ASE or ablating HSN is shown in Figure 6A. Figure 6A also demonstrates expression of a neuronal within the gonads upon disruption of the communication within the neuronal circuit by interfering with the production of the neuropeptide FLP-6 and of the neurotransmitters serotonin and Ach.

2. The reviewers agreed that more evidence is needed to demonstrate that flp6 is a target of RIDD. The suggestions for experiments to support this conclusion are as follows: (a) measure flp6 RNA levels upon inducing ER stress, and when Ire1 is expressed in ASE, with the prediction that flp6 RNA levels should decrease;

We now show by qRT-PCR that *flp-6* RNA levels are reduced under different ER stress conditions. Figure 3B shows that *flp-6* RNA levels are reduced by *xbp-1* depletion, in an *ire1*-dependent manner. Note that *xbp-1* and *ire-1* deficiencies by themselves induce ER stress. Figure 3-supp 2A shows that *flp-6* RNA levels are reduced upon ER stress induced by *tfg-1* RNAi (in an *ire-1*-dependent manner). Note that *tfg-1* RNAi induces ER stress. Figure 3C shows *that flp-6* RNA levels are reduced by *ire-1* overexpression in the ASE neuron (note that *ire-1* overexpression is enough to drive its activation).

(b) if possible (given potential difficulties with the Xbp1 mutants), demonstrate that flp6 RNA degradation is independent of Xbp;

We now show in Figure 3B that *flp-6* RNA levels are reduced in *xbp-1* mutants, but not in *ire-1; xbp-1* double mutants. Note that *xbp-1* and *ire-1* deficiencies by themselves induces ER stress, but only the xbp-1 deficiency destabilizes *flp-6* RNA.

(c) indicate whether the in vitro cleavage assay results in cleavage at the putative cleavage site described by the authors, and test whether mutation of this site prevents degradation of the flip6 RNA in vitro and in vivo.

In-vitro: “Indeed, incubation of the flp-6 stem-loop RNA fragment with purified recombinant IRE-1 KR-3P generated clearly detectable cleavage products of the expected fragments (Figure 3E).” – Page 11. Furthermore introduction of a scrambled sequence instead of the sequence encoding the conserved loop motif diminished the in-vitro cleavage of the transcript by IRE1 in vitro (Figure 3 - sup3B).

In-vivo: We have also added experiments to show that *ire-1's* effects on GED are mediated primarily through FLP-6. This has been achieved by generating CRISPR-designed worms harboring silent mutations in the *flp-6* gene that disrupt the sequence and structure of the predicted cleavage site while preserving the CDS. We find that this mutation stabilizes the *flp-6* transcript under ER stress conditions and protects from ER stress-induced GED in otherwise WT animals.

3. The authors should test the specificity of serotonin for the ASE-AIY-HSN-germline circuit. Can the effects of tgf-1 RNAi on GED also be blocked by serotonin? Does serotonin interfere with the mex-3 RNAi GED phenotype?

We demonstrate the specificity of serotonin for the ASE-AIY-HSN-germline circuit-induced GED. We show that serotonin supplementation blocks the GED induction by *flp-6* deficiency (Figure 5-supp2) or by *tfg-1RNAi* (i.e. ER stress) -induced GED (Figure 5D), but does not interfere with the *mex-3* RNAi GED phenotype (Figure 5D).

4. Given the potential therapeutic aspect of this work and the previous findings that Ire1 activation allows animals with tumors to live longer, the authors should test how the flp6 mutant affects survival and/or paralysis in gld-1-RNAi treated animals.

Our previous work showed that IRE-1/ER stress-induced transdifferentiation of the tumorous germline in *gld-1(-)* animals limits the expansion of tumor. In Figure 6C,D we now show that *flp-6* deficiency, which induces transdifferentiation of the tumorous germline, also improves survival and reduces paralysis in *gld-1*-RNAi treated tumorous animals.

Reviewer #1 (Recommendations for the authors):My main suggestion for an aspect of this paper that could be strengthened is included in my public review. To address this, I would suggest that the authors determine whether flp6 degraded when Ire1 is specifically expressed in the ASE as in figure 3H.

We now show in Figure 3C that over-expression of an *ire-1* transgene specifically in the ASE neuron under the *che-1P* reduces the levels of the *flp-6* transcript as assessed by qRT-PCR.

A second suggestion is about the in vitro assay. The authors also show an experiment where Ire1 appears to cleave the FLP6 RNA in Figure 3D. To further support that this is a specific cleavage event, it would be nice to know if the sizes of the cleavage products in Figure 3D are consistent with cleavage at the stem loop structure described by the authors. Even more conclusive would be if the authors showed that mutating the critical nucleotide(s) in the cleavage sites prevented cleavage, or that a second RNA without such a site is not cleaved in this assay.

This is a great suggestion! Indeed, the sizes of the cleavage products in Figure 3E are consistent with cleavage at the predicted stem loop structure. Furthermore introduction of a scrambled sequence instead of the sequence encoding the conserved loop motif diminished the in-vitro cleavage of the transcript by IRE1 in vitro (Figure 3 – sup3B).

Reviewer #2 (Recommendations for the authors):1. A central point of the article is to show for the first time a RIDD target in *C. elegans*. Although their findings are consistent with this conclusion, more evidence is required to firmly establish flp-6 as a RIDD target. Measuring the levels of flp-6 in other ER stress conditions, such as tunicamycin treatment, would help to build a convincing case that flp-6 is a genuine RIDD target.

*flp-6* levels were reduced by several different ER stress conditions:

– *ire-1*-dependent downregulation of the *flp-6::gfp* translational reporter upon tunicamycin treatment is shown in Figure 3D.

– *ire-1*-dependent downregulation of the *flp-6::gfp* translational reporter upon induction of ER stress by *tfg-1* RNAi treatment is now shown in (Figure 3-sup2C).

– *ire-1*-dependent downregulation of the *flp-6* RNA upon induction of ER stress by *tfg-1 RNAi* treatment is now shown in (Figure 3-sup2A).

– *ire-1*-dependent downregulation of the *flp-6* RNA upon induction of ER stress by *xbp-1* deficiency is shown in (Figure 3B).

The authors establish that flp-6 can act as a direct target of the IRE-1 endoribonuclease in vitro, but not that it is actually a direct target in vivo; mutating the predicted stem loop structure and demonstrating an inhibition of flp-6 loss and downstream effects on GED would solidify the authors argument.

This is a great suggestion! As requested, we mutated the predicted stem loop structure (altering the RNA sequence using silent mutations that preserve the coding sequence). We confirmed that these mutations stabilize the *flp-6* transcript under ER stress conditions (Figure 3F) and show that ER stress induced by *tfg-1* RNAi does not induce GED in these *flp-6* stem-loop mutants (Figure 3G and Figure 3 – supp3C).

It is also important to exclude any potential role of the XBP-1-mediated UPR in flp-6 loss, by measuring flp-6 levels in xbp-1(-) mutants – although this might be a difficult result to interpret as previous author data indicates that xbp-1(-) mutants already display a basal increase in the ER stress levels (Safra M et al., 2013; Levi-Ferber M et al., 2015).

We followed *flp-6* transcript levels using qRT-PCR and found that they decrease in *xbp-1(-)* background, and that this decrease is dependent on *ire-1* (Figure 3B).

Also, in addition to RIDD and xbp-1 splicing, IRE-1 has other functions in the cell, such as regulating JNK/TRAF2 signaling. Authors should exclude (or at least discuss) any potential role of JNK/TRAF2 in the regulation of GED by IRE-1.

We now note on page 18 of the discussion that:

“Whereas our findings clearly implicate RIDD in ER stress-induced GED, this study has not explored the possible implication of additional *xbp-1*-independent signaling pathways regulated by IRE-1 such as the *JNK*/*TRAF2* pathway.”

2. The authors did a fantastic job in mapping the neuronal circuits controlling GED. Testing whether the effects of tgf-1 RNAi on GED can also be blocked by serotonin would help to support the model proposed in Figure 6. In addition, checking whether serotonin interferes with the mex-3 RNAi GED phenotype would help demonstrate serotonin specificity for the ASE-AIY-HSN-germline circuit.

This is a great suggestion! We now show in Figure 5D that serotonin supplementation indeed blocks both ER stress-induced GED and *flp-6* deficiency-induced GED, but does not affect mex-3-induced GED. Thus serotonin seems to regulate GED specifically in the context of the ASE-AIY-HSN-germline circuit.

3. The authors emphasize in the discussion the potential therapeutic aspect of the work. In their previous paper, they were able to show that IRE-1 activation allows animals with tumors to live longer (Levi-Ferber et al., 2015). To further support the therapeutic aspect of this work, I suggest that the authors test the effects of flp-6 mutant in survival and/or paralysis in gld-1-RNAi treated animals.

Our previous work showed that IRE-1/ER stress-induced transdifferentiation of the tumorous germline in *gld-1(-)* animals limits the expansion of tumor. We now show that *flp-6* deficiency, which also induces transdifferentiation of the tumorous germline, improves survival and reduces paralysis in *gld-1-*RNAi treated tumorous animals.

Reviewer #3 (Recommendations for the authors):Figure 1: The rrf-1(pk1417) I used, according to the publications cited by the authors, can undergo at least partial RNAi mediated knockdown in the intestine, and show elevated expression of, at least some, transgenic arrays.

We state in the text on page 5 that:

“we used mutants in the *rrf-1* gene, whose RNAi activity is compromised in most somatic tissues but whose germline RNAi activity is intact”.

Since we found that *tfg-1* RNAi treatment failed to induce GED in the *rrf-1* background, we can safely conclude that *tfg-1* RNAi treatment in the germline (and in whatever somatic tissues that still partially respond to the RNAi treatment in this background) is not sufficient to induce GED. Hence a wider RNAi response is required for the induction of the GED phenotype. Had the result been that *tfg-1* RNAi treatment was sufficient to induce GED – then it would have been important to further dissect whether the phenotype is due to RNAi responsiveness in the germline or in the weakly responding somatic tissues – however this was not the case.

The conclusion that the critical site of action of ER stress in this case is the soma and not the germline is further supported by a wide-range of IRE-1-rescue experiments demonstrating that rescue of IRE-1 expression in the soma/all neurons/sensory neurons/ASE are each sufficient to support ER stress-induced GED.

In addition, other artifacts of triple RNAi have not been excluded in the studies shown here.

The effectiveness of triple RNAi has been confirmed in several ways:

1) Experimentally – based on the phenotypes of the germline tumor (*gld-1* RNAi), reduced animal size (*tfg-1* RNAi) and lack of apoptosis (SYTO12 staining of *ced-3* RNAi animals).

2) As shown in Figure 1 sup 2 in Levi-Ferber M et al., 2015, western blot has also confirmed the efficiency of the *gld-1* RNAi treatment in the context of single/double/triple RNAi experiments.

3) Triple RNAi was used only in the beginning of the paper (Figure 1 and some of Figure 2). After that, the experiments were done in a less complex setting in which one of the RNAi treatments was omitted and/or replaced by mutations, giving similar results. Specifically, *ced-3* RNAi was replaced by a *ced-3(n1286)* mutation in Figure 1D, Figure 1F, Figure 5D, Figure 6A. In many experiments (Figure 2E and onwards) the neuronal circuit was disrupted via various mutations and the animals were treated only with a mixture of *ced-3* and *gld-1* RNAi.

The images are low resolution, and the germline looks largely disturbed and DAPI is used as a proxy for GED.

Figure resolution may have dropped in the PDF format. Original figures of higher resolution are now attached. Note that the analysis is done at day 3-4 of adulthood. The germline at this time point is not as nice as in day1 animals. The issue of using DAPI as a proxy for GED has been addressed by adding experiments showing the expression of a neuronal marker in the gonad, and by referring to GED only after seeing this expression. Otherwise, we clarify that the aberrant DAPI-stained nuclei are only a proxy for GED.

While it is possible some of the cells are differentiating ectopically (vis-a-vie the authors' previous publication), that needs to be shown.

We have added experiments using somatic markers to confirm the induction of somatic cells in the gonads have been added:

Figure 1D shows neuronal cells filling the gonads upon neuronal over-expression of *ire-1.*

Figure 2F shows induction of neuronal and muscle cells within the gonad upon *flp-6* RNAi treatment.

Figure 5D shows neuronal cells filling the gonads upon *tfg-1, flp-6* or *mex-3* RNAi treatment.

Figure 6 shows neuronal cells filling the gonads upon disruption of the ASE-AIY-HSN neuronal circuit using a variety of mutations (che-1, flp-6, cha-1, sem-4 and tph-1).

Authors should not use 'abnormal nuclei' interchangeably with GED (eg. nuclei in tph-1 mutants in Figure 5)

We are now careful to use the term 'abnormal nuclei' when referring to the DAPI staining results, and GED when referring to the more direct experiment using the neuronal marker to demonstrate the somatic nature of the observed cells.

IRE-1 is supposed to be required for GED. However, Figure 1B, tgf-1 (RNAi); ire-1 (-) animals appear to have GED.

It is correct that IRE-1 is supposed to be required for GED. In Figure 1B, *tfg-1 (RNAi); ire-1 (-)* animals do not have GED. It may have appeared so due to the high density of the tumorous germline in the gonad. We replaced the specific picture with a clearer one. As noted in the figure legend, we analyzed hundreds of *ire-1(-)* animals for this phenotype, and they do not display ER stress-induced GED.

The effects of IRE-1 manipulation on ER stress needs to be carefully assayed using qRT-PCR, XBP-1 splicing etc., before one can conclude, as the authors do that " active IRE-1 signaling rather than ER stress" modulates GED. In this regard, it would be important for the authors to spell out what they mean by ER stress. Where? Which sensors etc.

The statement that "active IRE-1 signaling rather than ER stress" modulates GED refers specifically to the experiments in which overexpression of IRE-1 was sufficient to induce GED. We now added a reference of our published work showing that overexpression of the *ire-1* transgene in *C. elegans* drives its activation as assessed by the induction of *xbp-1* splicing.

Controls for o/e of ire-1 in each of the tissues alone should be shown.

The tissue-specific expression of *ire-1* in the different tissues has been achieved by driving its expression using commonly used tissue specific promoters (*Prgef-1, Pmyo-3 and Pges^-1^*). Relevance of specific neurons to GED induction has been further demonstrated using mutants that interfere with the identity/pattern of specific neurons, with the production of their communication molecules.

Figure 2: Previous studies by other groups have shown that overexpressing spliced XBP-1 in neurons of *C. elegans* induced ER stress in other tissues (shown in intestine) through neurotransmitter release (unc-13 mechanism). Even though the authors believe that the phenomenon they are studying is XBP-1 independent, and dependent on IRE-1 RIDD activity, the use a mutant IRE-1 unable to splice XBP-1 would be the conclusive way to demonstrate this. In its absence, the observations could be due to IRE-1 induced XBP-1 splicing.

This is an excellent suggestion, however there is no good way to manipulate *ire-1* in *C. elegans* such that *xbp-1* splicing is prevented while RIDD is preserved. Thus the only way to eliminate the role of *xbp-1* in the ER-stress induced GED process is to conduct the experiments *in xbp-1(-)* animals. Indeed, in our 2015 *eLife* paper which is the basis for the current paper, we clearly demonstrate that ER-stress induced GED occurs independently of *xbp-1*. Thus, *xbp-1* (spliced or unspliced) is not required for this phenomenon to occur.

Figure 3: It is unclear why the ire-1(-) animals not show the same phenotype as flp-6 (-) animals? Without flp-6, GED is constitutive (in a gld-1, ced-3 background). Without ire-1 GED is inhibited (in same background). Yet in a flp-6; ire-1(-) strain GED is activated similar to flp-6 null-the reason for this is not clear.

Our suggested model is that *ire-1* promotes ER-stress induced GED by destabilizing the *flp-6* transcript, which encodes a protein (FLP-6) that prevents GED. Thus, the consequence of ER stress is *ire-1* activation and consequently *flp-6* depletion (or at least reduction in *flp-6* levels). In the absence of *ire-1, flp-6* transcript levels remain high in spite of the ER stress (as there is no IRE-1 to perform RIDD) and therefore GED is inhibited. In the *flp-6(-)* background, there is no need for *ire-1*-mediated removal of *flp-6* as there is no *flp-6* to begin with, therefore GED cannot be prevented under these conditions (and it even does not require ER stress).

How do authors reconcile ire-1 independence of GED in flp-6 mutants with an ire-1 dependent hsp-4::GFP is a transgene that is expressed mainly in intestine, and as such is an incomplete readout of ER stress. Therefore, its sole use to establish that flp-6(-) animals do not activate ER stress is not supported.

The *hsp-4::gfp* transgene is widely accepted as a marker for activation of the *ire-1/xbp-1* arm of the UPR. I agree that it is predominantly observed in the intestine and the spermatheca, however, we now specifically analyzed its expression is ASE (where it is expressed, although not the main site of expression). We focused on ASE since our data indicate that it is the cell in which ire-1 is required to promote ER stress-induced GED.

We agree that while the lack of *hsp-4* induction is consistent with the idea that the IRE-1/XBP-1 arm of the UPR is not hyperactivated in these animals, it could be that it is an irrelevant marker for ER stress in this case. That is why we also asked if the observed GED in *flp-6(-)* animals is *ire-1* dependent. The idea was that if *flp-6* deficiency induces ER stress which then leads to GED, then GED induction should be *ire-1* dependent (as ER stress-induced GED is *ire-1* dependent). However, we found that *ire-1* is not required for GED induction upon *flp-6* deficiency. Hence ER stress does not act downstream to *flp-6* deficiency to induce GED.

Neuronal expression of the Zcls4s has not been previously established. Images of expression that was quantified in ASE should therefore be shown.

Please see Author response image 1.

**Author response image 1. sa2fig1:** 

Is the FLP-6 translational reporter expression levels restored in the wy762 mutant?

We only looked at the dependency of ER stress-induced GED on the *wy762 ire-1* mutant (Figure 3H).

The analysis of the FLP-6 translational reporter was done in the *ire-1(ok799)* null background (Figure 3D and Figure 3 – sup2c).

IRE-1 RIDD activity has not been demonstrated in *C. elegans*, and if FLP-6 is indeed an IRE-1 RIDD target, this would be exciting new advance in the field. Therefore these data while provocative, should be validated in independent ways. A convincing method to show IRE-1 RIDD activity would be to re-express an IRE-1 kinase domain only (or mutant without nuclease activity) to see if does not decrease expression of the flp-6 translational reporter. For instance, does is flp-6 lacking the stem loop also a RIDD substrate in the in vitro assays? What is the efficacy of flp-6 processing the RIDD reaction ? How well would known RIDD substrates be cleaved in this assay? If ER-UPR is activated in the FLP-6 translational reporter could be downregulated and explain the results.

As discussed above, we provide more in-vivo and in-vitro evidence demonstrating that flp-6 is a target of RIDD.

*In-vitro:* We confirmed that the in vitro cleavage assay results in cleavage products of the expected sizes, and that mutation of the predicted cleavage site prevents degradation of the flp-6 RNA. The in-vitro cleavage of the *flp-6* RNA substrate is detectable in this assay, although it is a weak substrate. This may be due to a variety of reasons such as a limiting co-factor or cross-species compatibility issues (the IRE1-KR domain is of human origin and the RNA to be cleaved is from *C. elegans*). It could also be that the recognition motif is not perfect compared to conventional mRNAs. Nevertheless, some cleavage of the substrate is observed, it is of the predicted size and it is dependent on the consensus loop sequence and hairpin. Furthermore, the *in-vivo* experiments following the endogenous levels of the *flp-6* transcript support the *in-vitro* results.

*In-vivo*: We now show that *flp-6* RNA levels are reduced under different ER stress conditions in an *ire-1*-dependent *xbp-1* independent manner. We show that over-expression of IRE-1 in ASE is sufficient to reduce *flp-6* transcript levels. We show that CRISPR-designed worms harboring silent mutations in the *flp-6* gene, that disrupt the predicted cleavage site while preserving the coding sequence, protect the stability of the transcript under ER stress conditions and protect the animals from ER stress-induced germline differentiation.